# Cell-selective proteomics reveal novel effectors secreted by an obligate intracellular bacterial pathogen

Allen G. Sanderlin [1], Hannah Kurka Margolis [1], Abigail F. Meyer[1] & Rebecca L. Lamason [1] ✉

Pathogenic bacteria secrete protein effectors to hijack host machinery and remodel their infectious niche. *Rickettsia* spp. are obligate intracellular bacteria that can cause life-threatening disease, but their absolute dependence on the host cell has impeded discovery of rickettsial effectors and their host targets. We implemented bioorthogonal non-canonical amino acid tagging (BONCAT) during *R. parkeri* infection to selectively label, isolate, and identify effectors delivered into the host cell. As the first use of BONCAT in an obligate intracellular bacterium, our screen more than doubles the number of experimentally validated effectors for the genus. The seven novel secreted rickettsial factors (Srfs) we identified include *Rickettsia*-specific proteins of unknown function that localize to the host cytoplasm, mitochondria, and ER. We further show that one such effector, SrfD, interacts with the host Sec61 translocon. Altogether, our work uncovers a diverse set of previously uncharacterized rickettsial effectors and lays the foundation for a deeper exploration of the host-pathogen interface.

*Rickettsia* spp. are Gram-negative bacteria that live exclusively inside eukaryotic host cells. Members of this genus include arthropod-borne pathogens that cause typhus and spotted fever diseases in humans and pose a significant global health risk[1,2]. By virtue of their intimate connection with the intracellular niche, these bacteria are poised to exploit host cell biology. *Rickettsia* spp., like other intracellular pathogens, secrete protein effectors to subvert diverse host cell processes, but their obligate intracellular lifestyle has precluded a detailed investigation of the host-pathogen interface[3]. Identifying such effectors and their host cell targets is an essential first step towards a mechanistic understanding of rickettsial biology and pathogenesis.

Extensive efforts to characterize proteins secreted to the rickettsial surface have revealed unique ways that these bacteria interact with host cell machinery. For example, the major outer membrane proteins OmpA and OmpB mediate host cell invasion[4,5], and the surface proteins Sca2 and RickA polymerize actin to drive motility within the host cytoplasm[6]. Furthermore, biochemical studies have identified myriad surface proteins that could likewise support the rickettsial life cycle[7–11]. Despite these advances, however, the subset of secreted proteins that *Rickettsia* spp. deliver into the host cell to drive infection has remained elusive; recent studies have characterized only a handful of such secreted rickettsial factors. For example, the effector Sca4 inhibits host vinculin and promotes rickettsial cell-to-cell spread[12]. RARP-2, a predicted protease, disrupts the *trans*-Golgi network during infection[13,14]. Moreover, the phospholipases Pat1 and Pat2 may both mediate escape from membrane-bound vacuoles[15,16], whereas Risk1 and RalF directly and indirectly manipulate host membrane phosphoinositides[17–19]. Aside from these six experimentally validated effectors, however, the effector arsenals of *Rickettsia* spp. remain a mystery. Given that other bacterial pathogens secrete dozens if not hundreds of effectors into the host cell[20–24], there is a pressing need to identify new rickettsial effectors.

An expanding suite of biochemical, genetic, and in silico methods has facilitated the identification of secreted effectors in a variety of bacterial pathogens. For example, effectors have been identified from bacteria grown in broth by fractionation and proteomic analysis[25–27].

[1]Department of Biology, Massachusetts Institute of Technology, Cambridge, MA, USA. ✉e-mail: rlamason@mit.edu

Reporter fusion libraries have enabled large-scale screens for secreted proteins[28,29], and heterologous expression by surrogate hosts has provided support for putative effectors of genetically intractable bacteria[30–32]. Computational tools, used in parallel with the above strategies, have highlighted core features of verified effectors to identify new candidate effectors[24,33].

However, reappropriating these methods for the discovery of rickettsial effectors remains a challenge. Axenic culture of *Rickettsia* spp. is not yet possible[34], and thus biochemical identification of secreted effectors must contend with overwhelmingly abundant host material[35]. Likewise, scalable reporter screens are limited by the inefficient transformation of these bacteria[3,34]. The short list of experimentally validated rickettsial effectors has hindered in silico identification of new candidates, especially if they lack the sequence features found in the larger effector repertoires of well-studied bacteria[36]. Heterologous expression bypasses these obstacles, but the secretion of a candidate effector ex situ does not prove its secretion during rickettsial infection. Through two-hybrid and co-immunoprecipitation approaches[13,17,18], a series of *rvh* effector molecules (REMs) has been identified based on interactions with the Rickettsiales *vir* homolog (*rvh*) type IV secretion coupling protein RvhD4[37]. Bioinformatic analyses have highlighted additional candidate REMs by virtue of their similarity to existing REMs, but secretion for many of these proteins has not yet been experimentally validated. Furthermore, interactions with RvhD4 are not conclusive proof of secretion because many of the other proteins that co-immunoprecipitate with RvhD4 include housekeeping proteins that are presumably not secreted into the host cell[17]. Unfortunately, the lack of a secretion-null *Rickettsia* mutant precludes validation of any effector as a true *rvh* substrate. Thus, alternative approaches are necessary to identify new secreted effectors.

Labeling strategies that enable the isolation of secreted effectors from the host cell milieu may circumvent these issues while remaining secretion system-agnostic. For example, bioorthogonal non-canonical amino acid tagging (BONCAT) permits metabolic labeling of newly synthesized proteins with amino acid analogs[38]. Labeling is restricted to cells expressing a mutant methionyl-tRNA synthetase (MetRS*) which, unlike the wild-type synthetase (WT MetRS), can accommodate the azide-functionalized methionine analog azidonorleucine (Anl)[39]. Anl-labeled proteins are then chemoselectively tagged with alkyne-functionalized probes by click chemistry for visualization or pull-down followed by mass spectrometry. This approach has been adapted to a variety of bacterial pathogens, including *Salmonella typhimurium*[40], *Yersinia enterocolitica*[41], *Mycobacterium tuberculosis*[42], and *Burkholderia thailandensis*[43], enabling selective labeling and isolation of bacterial proteins during infection.

We therefore implemented cell-selective BONCAT during infection with the obligate intracellular bacterium *Rickettsia parkeri*. Using this approach, we detected both known and novel secreted effectors, including proteins of unknown function found only in the *Rickettsia* genus. In addition to confirming their secretion, we demonstrate diverse localization patterns for these new effectors. Moreover, we show that the secreted effector SrfD localizes to the endoplasmic reticulum (ER) where it interacts with the host Sec61 complex. Our findings expand the toolkit for exploring rickettsial biology, which will provide much-needed insight into how these pathogens engage with the host cell niche.

## Results

### BONCAT permits selective labeling of rickettsial proteins

We sought to identify new effectors secreted during rickettsial infection. We needed an approach that would overcome the limitations associated with the rickettsial lifestyle and enable the detection of low abundance effectors in the host cytoplasmic milieu[35]. Inspired by the use of cell-selective BONCAT with facultative intracellular bacteria, we adapted this technique to the obligate intracellular bacterial pathogen, *R. parkeri*, to label rickettsial proteins for subsequent identification (Fig. 1a). We first generated *R. parkeri* harboring a plasmid encoding MetRS*. To determine if MetRS* expression adversely impacted rickettsial infection, we performed infectious focus assays in A549 host cell monolayers. We found that infectious foci formed by the MetRS* strain were indistinguishable in both size and bacterial load from those formed by the WT strain (Fig. 1b, c), indicating that MetRS* expression does not impede cell-to-cell spread or bacterial growth, respectively.

Having confirmed that *R. parkeri* tolerates MetRS* expression, we next tested the functionality of MetRS* to label rickettsial proteins. We infected A549 cells for two days and then treated infected cells with Anl for 3 h prior to fixation. To visualize the incorporation of Anl by fluorescence microscopy, we tagged labeled proteins with an alkyne-functionalized fluorophore. As expected, labeling was restricted to MetRS* bacteria following treatment with Anl (Fig. 1d). To evaluate labeling of secreted and non-secreted proteins during infection, we used a previously established selective lysis protocol to separate the infected host cytoplasm from intact bacteria after 3 h of Anl labeling[44]. We then tagged labeled proteins from each fraction with alkyne-functionalized biotin and detected them by Western blotting. Consistent with our microscopy results, only the MetRS* strain exhibited appreciable labeling following treatment with Anl (Fig. 1e). Within this Anl-labeled, MetRS*-infected sample, the pellet fraction yielded a smear of bands, as expected for proteome-wide incorporation of Anl. Furthermore, the supernatant fraction contained several unique bands not found after infection with WT bacteria similarly treated with Anl (Fig. 1e, lane 4 versus lane 2). Altogether, these findings demonstrate that BONCAT can be used to selectively label proteins produced by obligate intracellular bacteria during infection.

### BONCAT identifies known and novel secreted effectors

We hypothesized that the unique bands present in the supernatant fraction during infection with MetRS* bacteria represented secreted rickettsial effectors. To identify these effectors, we infected cells for two days, labeled with Anl during the last 5 h of infection, tagged cytoplasmic fractions with alkyne-functionalized biotin as before, and isolated biotinylated proteins using streptavidin resin. We then analyzed these pull-downs by mass spectrometry to identify rickettsial proteins (Fig. 2, Supplementary Data 1).

This analysis yielded twelve hits, several of which had been previously studied. Importantly, these included proteins previously characterized as secreted effectors, providing validation of our approach. We identified the patatin-like phospholipase A₂ enzyme Pat1[16], the ankyrin repeat protein RARP-2[13], and the phosphatidylinositol 3-kinase Risk1[17], all known secreted effectors. The autotransporter proteins Sca1 and OmpA were also identified in the supernatant fraction despite their localization to the bacterial outer membrane[45]. However, both Sca1 and OmpA are post-translationally processed[46–48], and the tryptic peptides from our experiments mapped exclusively to their surface-exposed passenger domains (Supplementary Fig. 1), suggesting that we detected the cleavage-dependent release of surface proteins into the host cytoplasm. A high abundance of internal rickettsial proteins was not robustly detected with this approach[8], confirming that contamination of the supernatant fraction from adventitious bacterial lysis was minimal.

The remaining proteins identified in our screen include seven putative secreted rickettsial factors (SrfA−G). SrfA is a predicted *N*-acetylmuramoyl-L-alanine amidase and the *R. conorii* homolog RC0497 exhibits peptidoglycan hydrolase activity[49]. The hypothetical protein SrfD has partial sequence homology to uncharacterized pentapeptide repeat-containing proteins in diverse taxa, but only *Rickettsia* spp. encode homologs of full-length SrfD. The remaining Srfs are hypothetical proteins with no sequence homology outside the *Rickettsia* genus. For further insight into these hypothetical proteins, we used a

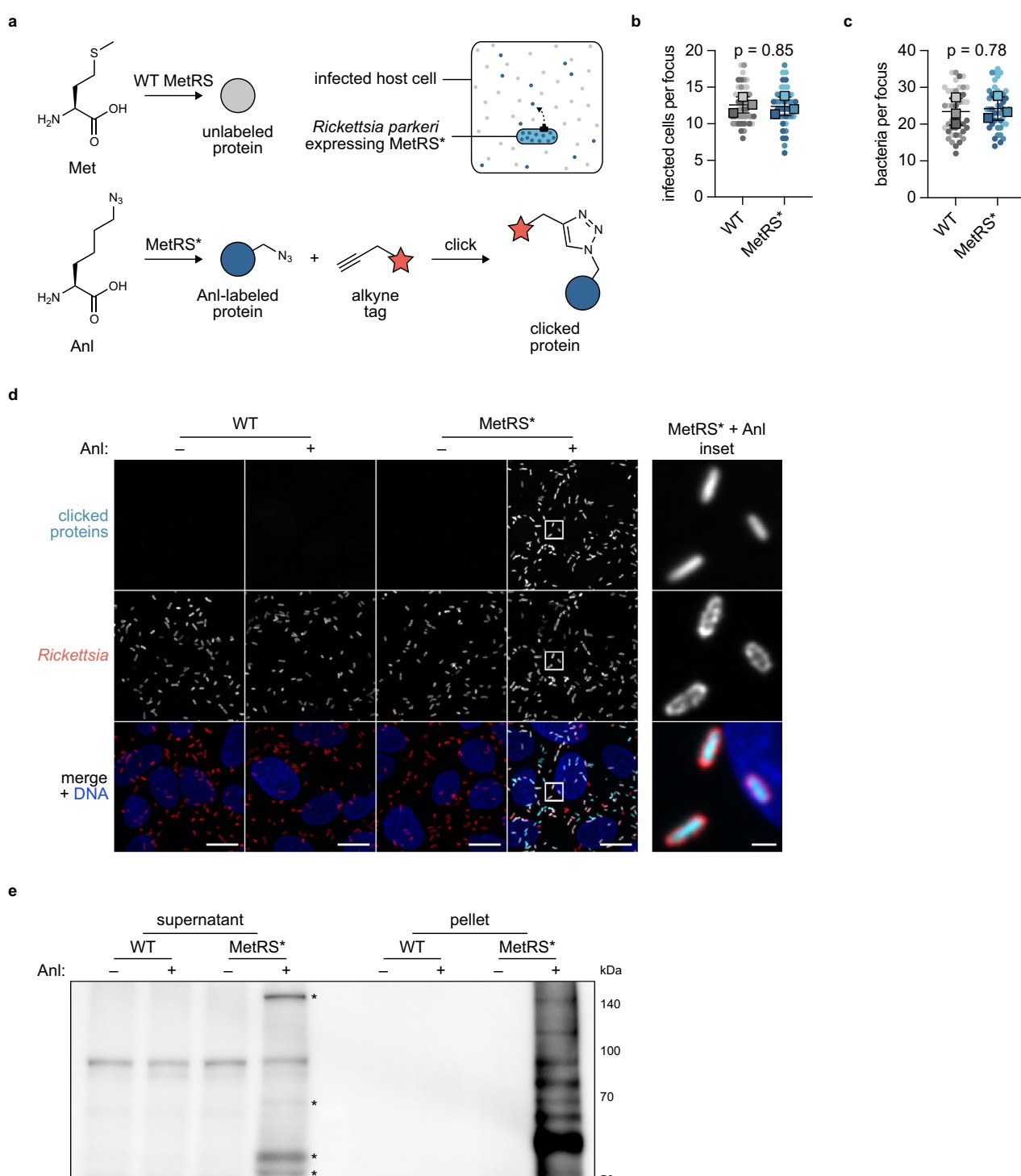

**Fig. 1 | BONCAT permits selective labeling of rickettsial proteins. a** Schematic of BONCAT approach. *Rickettsia parkeri* expressing a mutant methionyl-tRNA synthetase (MetRS*) incorporates the Met analog azidonorleucine (Anl) into nascent proteins, some of which are secreted into the host cell during infection. Anl-labeled proteins (blue circle), but not unlabeled proteins (gray circle), are tagged (red star) by click chemistry. Wild-type (WT) MetRS cannot accommodate Anl. **b** Infected cells and **c** bacteria per focus during infection of A549 cells by WT or MetRS* *R. parkeri*. The means from *n* = 3 independent experiments (squares) are superimposed over the raw data (circles) and were used to calculate the means ± SD and *p*-values (unpaired two-tailed *t*-test, *t* = 0.206 and 0.297, *df* = 4). Data are shaded by replicate experiments. **d** Images of WT and MetRS* *R. parkeri* treated with (+) or without (−) Anl during infection of A549 cells (Hoechst, blue). Bacteria were permeabilized and stained (red), and Anl-labeled proteins were detected by tagging with an alkyne-functionalized fluorescent dye (cyan). Scale bar, 10 μm (inset, 1 μm). **e** Western blot for biotin in lysates harvested from A549 cells infected with WT or MetRS* *R. parkeri* with (+) or without (−) Anl treatment. Infected host cells were selectively lysed to separate supernatants containing the infected host cytoplasm from pellets containing intact bacteria. Anl-labeled proteins were detected by tagging with alkyne-functionalized biotin. Asterisks indicate putative secreted effector bands. Results for (**d**) and (**e**) are representative of at least three independent experiments. Source data are provided as a Source Data file.

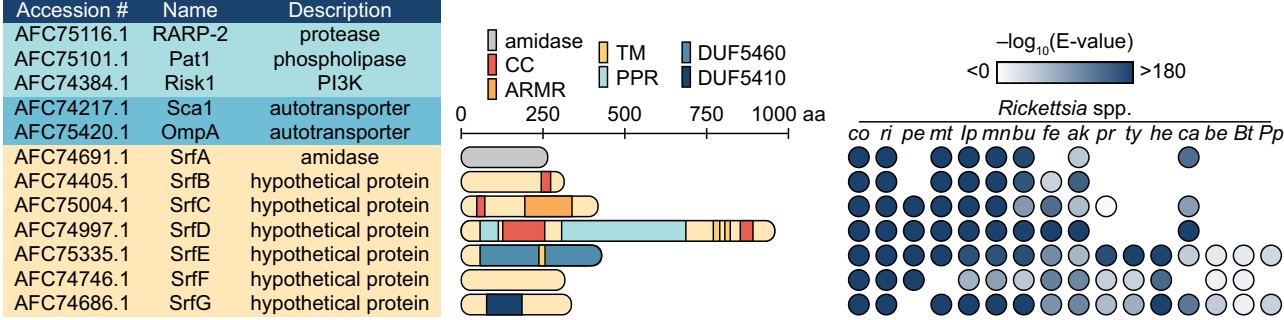

**Fig. 2 | BONCAT identifies novel secreted rickettsial factors (Srfs) that are structurally diverse and variably conserved.** *R. parkeri* protein hits identified from infected host cytoplasmic lysates using BONCAT include proteins that had been experimentally validated as secreted effectors in previous studies (light blue), autotransporter proteins (dark blue), and the novel secreted effectors SrfA–G (beige). Hit calling is described in the "Methods" section. Protein lengths, putative domains, and structural motifs are indicated for SrfA–G. CC coiled coil, ARMR armadillo-like repeats, TM transmembrane helix, PPR pentapeptide repeats, DUF Pfam domain of the unknown function. BLAST E-values were computed to evaluate the similarity between *R. parkeri* SrfA–G and putative homologs in select members of the *Rickettsia* genus. Species lacking a detected Srf homolog were left blank. *co R. conorii, ri R. rickettsii, pe R. peacockii, mt R. montanensis, lp Rickettsia* endosymbiont of *Ixodes pacificus, mn R. monacensis, bu R. buchneri, fe R. felis, ak R. akari, pr R. prowazekii, ty R. typhi, he R. helvetica, ca R. canadensis, be R. bellii, Bt Rickettsia* endosymbiont of *Bemisia tabaci* MEAM1, *Pp Rickettsia* endosymbiont of *Pyrocoelia pectoralis*. Source data are provided as a Source Data file.

variety of remote homology prediction tools to identify putative domains[50–53]. In addition to transmembrane helices, several Srfs are predicted to contain potential protein-protein interaction motifs, such as α-superhelical armadillo-like repeats, β-solenoid-forming pentapeptide repeats, and coiled coils[54–56]. Finally, SrfE and SrfG contain the *Rickettsia*-specific domains of unknown function DUF5460 and DUF5410, respectively. In a recent bioinformatic analysis[37], SrfG was nominated as a candidate REM (cREM-2b) but had not been validated as a secreted effector or RvhD4 interaction partner in that work.

The Srfs are variably conserved within the *Rickettsia* genus. For example, homologs of SrfE are found across the genus, whereas SrfB homologs are only present in a subset of species. Some Srf homologs are fragmented (e.g., *R. felis* SrfB) or otherwise highly divergent from the *R. parkeri* Srf (e.g., *R. typhi* SrfF), suggesting that Srf function is not strictly shared between all species. Furthermore, the presence or absence of a given Srf homolog appears to be independent of pathogenicity: pathogenic (cf., *R. rickettsii* str. Sheila Smith and *R. typhi*) and non-pathogenic (cf., *R. peacockii* and *R. buchneri*) species alike encode either full sets of Srf homologs or are missing particular Srfs.

Additionally, the *srf* loci are scattered across the *R. parkeri* genome (Supplementary Fig. 2a), in contrast to the effector gene clusters (pathogenicity islands) observed in more well-studied pathogens[57]. The fact that the *srf* loci are not obvious from studies of rickettsial genome architecture reinforces the value of experimentally identifying effectors secreted by these bacteria. Similar to known secreted rickettsial effectors (e.g., RARP-2), the Srfs are also not encoded proximal to components of either the type IV (T4SS: *rvhBD*) or type I (T1SS: *tolC, aprDE*) secretion systems, which may mediate Srf export to the host cell[58]. Moreover, in silico T4SS effector search tools do not clearly predict SrfA–G as likely effectors[36,59]. Similarly, SrfA–G lack the glycine-rich repeat motifs common in T1SS effectors[60]. The limitations of such bioinformatic methods for Srf identification underscore the utility of our proteomics-based approach to uncover putative rickettsial effectors.

The *srf* gene neighborhoods are largely conserved across the *Rickettsia* genus (Supplementary Fig. 2b), and the flanking genes are often intact even in species where a particular *srf* is fragmented or absent. Furthermore, with the exception of the *srfG* and *cREM-2a* gene pair encoding DUF5410-containing proteins[37], there is no obvious functional link between the *srf* genes and the conserved flanking genes. In contrast to their secreted effector neighbors, the proteins encoded by these flanking genes include those involved in housekeeping functions like DNA repair and recombination (e.g., XerD, RadA, and RecO), tRNA and rRNA modification (e.g., TsaB,

RluB, RsmD, and MnmE), translational initiation (InfB), and peptidoglycan processing (Slt and IdcA). Altogether, these findings motivate a more comprehensive analysis of *srf* evolution and diversification in future work.

## Srfs are secreted by *R. parkeri* into the host cell during infection

We next sought to confirm the secretion of SrfA–G by *R. parkeri* using a previously validated orthogonal approach. We generated *R. parkeri* strains expressing Srfs with glycogen synthase kinase (GSK) tags and infected Vero host cells. Upon secretion into the host cytoplasm, GSK-tagged proteins are phosphorylated by host kinases[61]. This well-established strategy does not require selective lysis, and secreted proteins can be detected by immunoblotting with phospho-specific antibodies[13,32,44,62,63]. As expected, a non-secreted control (GSK-tagged BFP) was not phosphorylated whereas a secreted effector control (GSK-tagged RARP-2) was phosphorylated (Fig. 3a). Importantly, the lack of phosphorylation for GSK-tagged BFP demonstrates that there is negligible release of non-secreted proteins into the host cytoplasm during infection for erroneous phosphorylation. We extended this analysis to our GSK-tagged Srf strains and confirmed secretion for the majority of the effectors: SrfA, SrfC, SrfD, SrfF, and SrfG. Despite similar strain growth and expression from a common promoter (*ompA*), the expression of these GSK-tagged constructs varied considerably, with SrfA having the most robust expression. Additionally, expression of GSK-tagged SrfB and SrfE was not detectable and we were therefore unable to verify their secretion in this assay (Supplementary Fig. 3). The strains expressing GSK-tagged SrfB and SrfE were GFP-positive and spectinomycin-resistant, indicating that they successfully maintained the expression plasmid. Nevertheless, the results from this assay demonstrate that the BONCAT screen revealed bona fide secreted effectors.

To confirm the secretion of the endogenous, untagged effectors, we raised antibodies against SrfC, SrfD, and SrfF. We then used selective lysis to check for secretion during infection of A549 host cells by WT *R. parkeri*. As shown previously[44], the bacterial RNA polymerase subunit RpoA was only detected in the pellet fraction, confirming that our selective lysis approach did not lead to adventitious rickettsial lysis that would confound validation (Fig. 3b). In contrast, we detected endogenous SrfC, SrfD, and SrfF in both the pellet and supernatant fractions, providing further validation that these effectors are secreted into the host cytoplasm. Given that GSK-tagged SrfE was not detectably expressed, we also raised antibodies against this putative effector. SrfE was present in both the pellet and supernatant fractions (Fig. 3c), confirming that the endogenous, untagged protein is

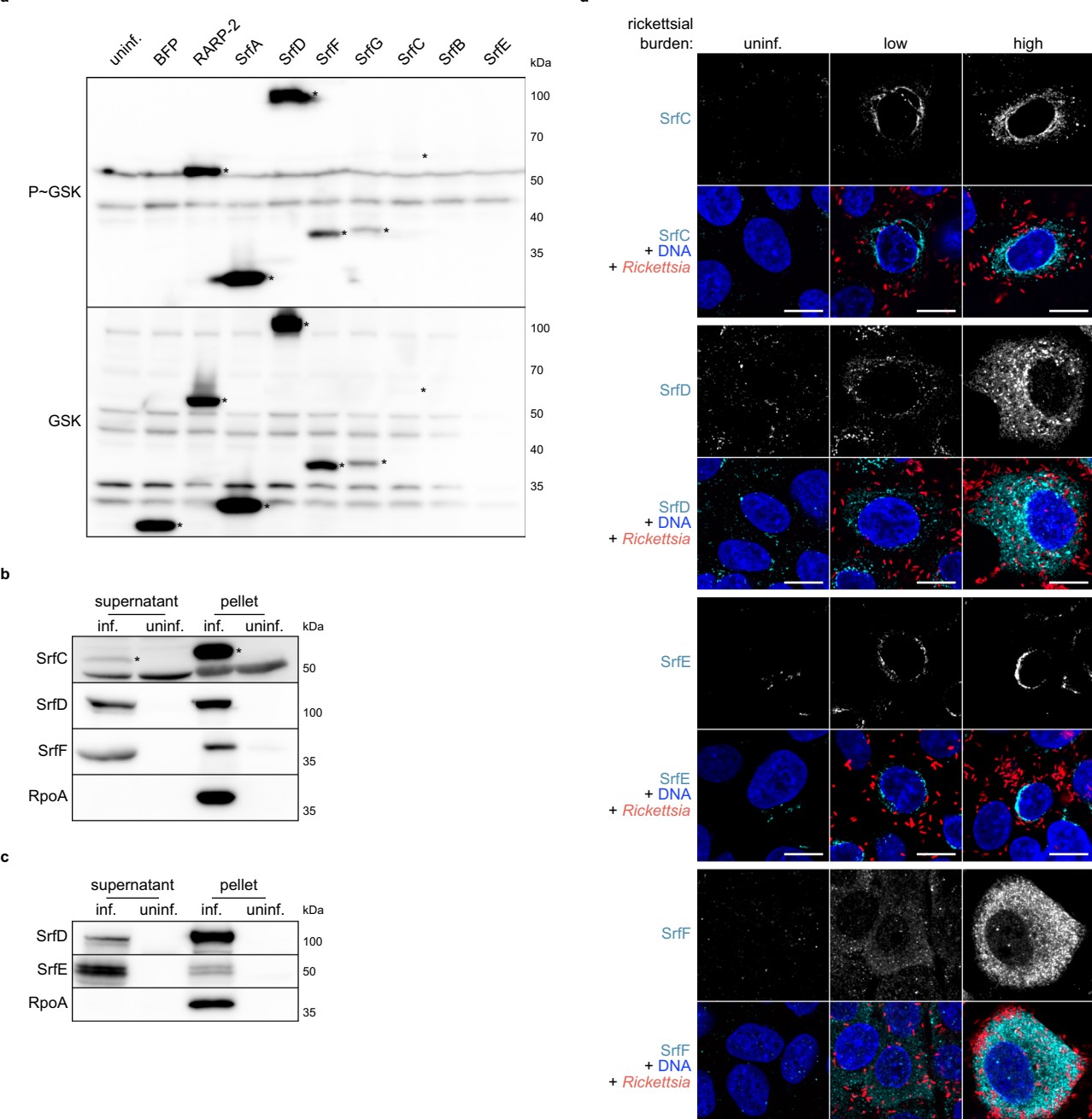

**Fig. 3 | Srfs are secreted by *R. parkeri* into the host cell during infection.**
**a** Western blots for GSK-tagged constructs expressed by *R. parkeri* during infection of Vero cells. Whole-cell infected lysates were probed with antibodies against the GSK tag (bottom) or its phosphorylated form (P-GSK, top) to detect exposure to the host cytoplasm. BFP (non-secreted) and RARP-2 (secreted) were used as controls. SrfA–G are ordered by observed expression level. Asterisks indicate GSK-tagged protein bands. SrfB and SrfE (expected 37 and 50 kDa, respectively) were not detected. **b** Western blots for endogenous, untagged SrfC, SrfD, and SrfF during *R. parkeri* infection of A549 cells. Infected host cells were selectively lysed to separate supernatants containing the infected host cytoplasm from pellets containing intact bacteria. Asterisks indicate SrfC bands (apparent 55 kDa, but expected 48 kDa). SrfD and SrfF ran at the expected sizes (107 and 36 kDa, respectively). RpoA, lysis control. **c** Western blots for endogenous, untagged SrfD and SrfE during *R. parkeri* infection of A549 cells. Lysates were prepared as in **b**. SrfE ran at the expected size (48 kDa). RpoA, lysis control. **d** Images of Srfs (cyan) secreted by GFP-expressing *R. parkeri* (red) during infection of A549 cells (Hoechst, blue). Srfs were detected at both low and high rickettsial burdens. Scale bar, 10 μm. Uninfected host cells (uninf.) were included as controls for (**a**–**d**), and the results are representative of at least two independent experiments. Source data are provided as a Source Data file.

delivered to the host cytoplasm despite the inconclusive GSK assay result. This finding underscores the importance of using multiple orthogonal approaches to validate effector secretion.

We next performed immunofluorescence microscopy to determine where endogenous, secreted SrfC, SrfD, SrfE, and SrfF localize during infection of A549 host cells (Fig. 3d). We observed rare instances (less than 3% of infected cells) of perinuclear staining for SrfC during infection, which was typically undetectable even at higher bacterial burdens. SrfE behaved similarly with rare instances (approximately 5% of infected cells) of perinuclear staining. For SrfD, we detected perinuclear speckles and faint diffuse staining that became more apparent with increased bacterial burden, possibly as a

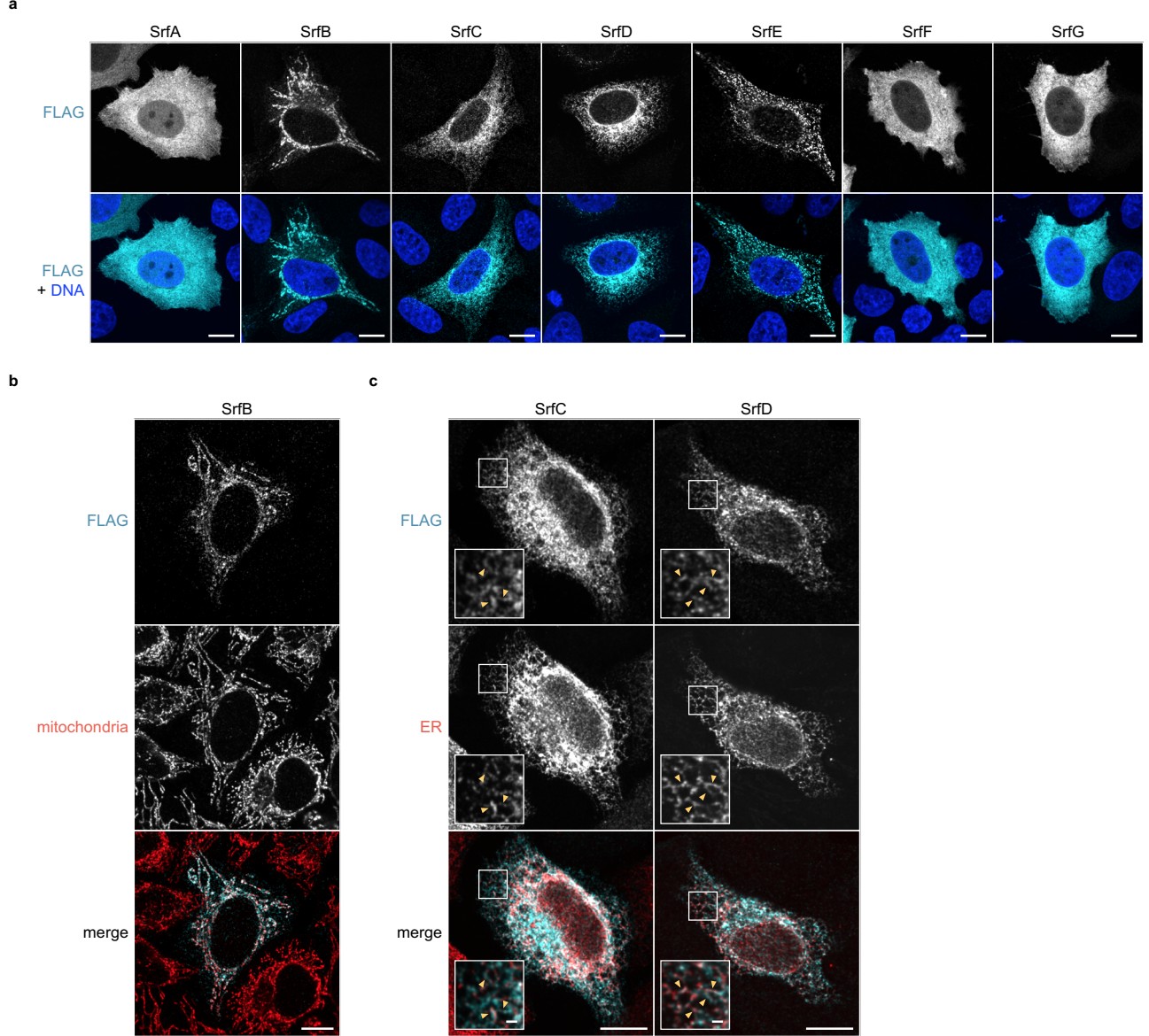

**Fig. 4 | Srfs exhibit diverse subcellular localization patterns. a** Images of 3xFLAG-tagged SrfA–G (cyan) expressed by transiently transfected HeLa cells (Hoechst, blue). Scale bar, 10 μm. **b** Images of 3xFLAG-tagged SrfB (cyan) in transiently transfected HeLa cells (mitochondrial AIF, red). White indicates an overlap between FLAG and AIF signals. Scale bar, 10 μm. **c** Images of 3xFLAG-tagged SrfC or SrfD (cyan) and ER-targeted mNeonGreen (red) expressed by transiently cotransfected HeLa cells. White indicates an overlap between FLAG and mNeonGreen signals. Scale bar, 10 μm (inset, 1 μm). Arrowheads highlight Srf colocalization with ER tubules. Results for (**a**–**c**) are representative of at least three independent experiments.

result of greater effector abundance. Finally, we noted staining for SrfF in the infected host cytoplasm, the intensity of which similarly increased at higher bacterial burdens. Our ability to detect each of these proteins in the host cell without bacterial permeabilization further demonstrates Srf secretion. Altogether, the results from multiple assays – selective lysis, reporter fusions, and microscopy-based approaches – confirm Srf secretion into the host cell.

## Srfs exhibit diverse subcellular localization patterns

Motivated by the varied staining patterns for SrfC–F during infection, we expanded our localization analysis to include the remaining Srfs. Secreted effectors target various subcellular compartments, and we reasoned that exogenous expression of these effectors in uninfected cells would offer a more tractable way to study their localization by microscopy[30,64]. We transiently expressed 3xFLAG-tagged SrfA–G in HeLa cells and used immunofluorescence microscopy to assess their localization (Fig. 4a). We observed diffuse staining of SrfA in the cytoplasm and nucleus. SrfB was detected along narrow structures of various sizes reminiscent of mitochondria. Colocalization between SrfB and mitochondrial apoptosis-inducing factor (AIF) confirmed this hypothesis (Fig. 4b), and we noted no obvious impact on mitochondrial morphology in SrfB-positive cells. SrfC and SrfD both exhibited a reticulate perinuclear localization pattern suggestive of localization to the ER, which was not as apparent for their endogenous, secreted counterparts detected during infection. Expression of SrfC or SrfD alongside ER-targeted mNeonGreen confirmed colocalization with ER tubules (Fig. 4c), and no obvious changes in ER morphology were noted for these cells. SrfE exhibited punctate staining throughout the cytoplasm, in contrast to the perinuclear staining noted for SrfE secreted during infection. Finally, we observed diffuse staining of SrfF and SrfG in the cytoplasm; for SrfF, this localization recapitulated the pattern we saw for the endogenous protein secreted during infection. Altogether, the diversity of these localization patterns suggests that the Srfs target distinct host cell compartments during infection.

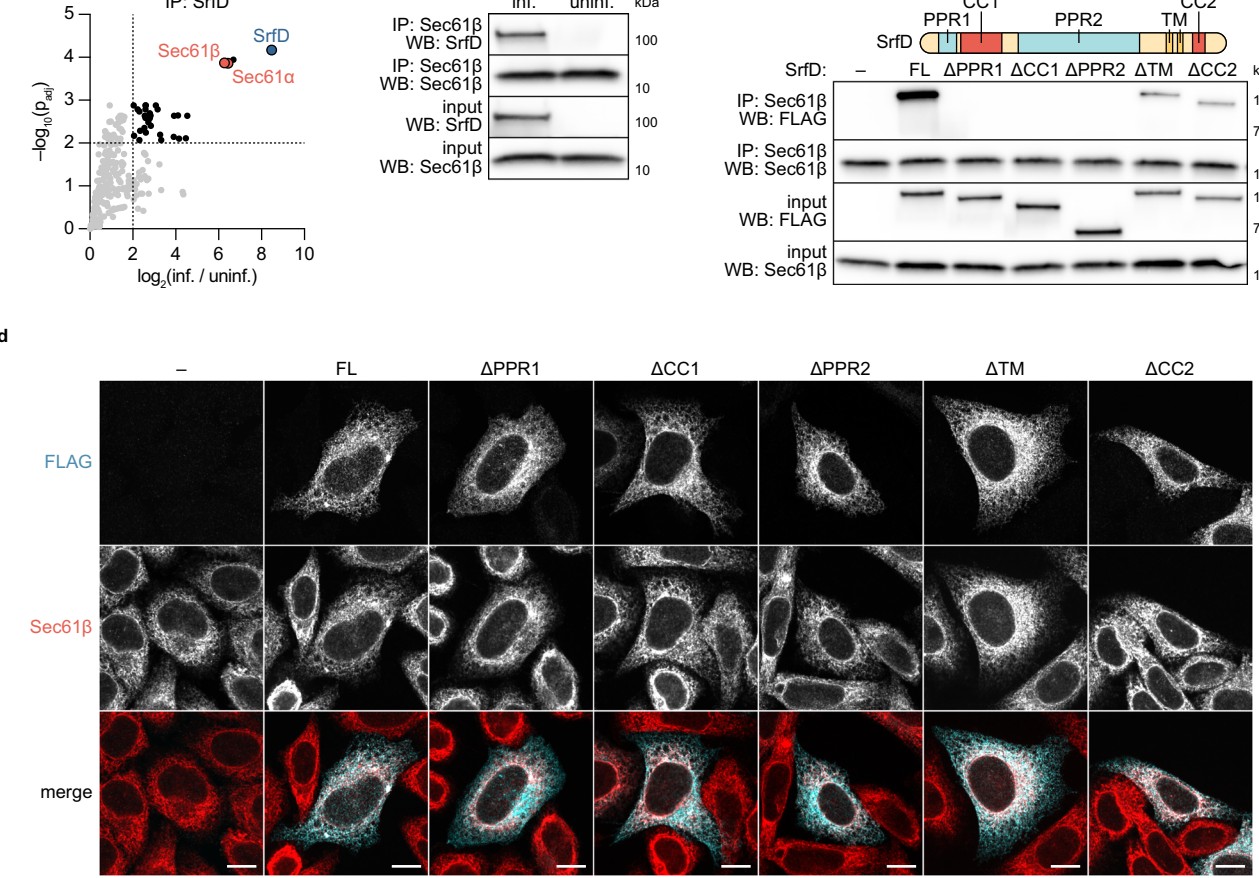

**Fig. 5 | SrfD interacts with host Sec61 and localizes to the ER via multiple domains. a** Proteins co-enriched following immunoprecipitation (IP) mass spectrometry of SrfD secreted by *R. parkeri* during infection of A549 cells. Protein abundance fold-changes and Benjamini–Hochberg adjusted *p*-values ($p_{adj}$) were computed for $n = 3$ independent infected (inf.) and uninfected (uninf.) samples (unpaired two-tailed *t*-test, $df = 4$). SrfD (blue), Sec61α/β (red), and thresholds for fold-change >4 and $p_{adj} < 0.01$ are indicated. **b** Western blots for SrfD and Sec61β following IP of Sec61β from A549 cells. Samples were prepared from *R. parkeri*-infected (inf.) and uninfected (uninf.) cells. **c** Western blots for FLAG and Sec61β following IP of Sec61β from HEK293T cells transiently transfected with 3xFLAG-tagged SrfD expression constructs. Putative domains and structural motifs are indicated. – empty vector, FL full-length SrfD. **d** Images of 3xFLAG-tagged SrfD constructs (cyan) from **c** expressed by transiently transfected HeLa cells (Sec61β, red). White indicates an overlap between FLAG and Sec61β signals. Scale bar, 10 μm. Results for (**b**–**d**) are representative of at least two independent experiments. Source data are provided as a Source Data file.

## SrfD interacts with host Sec61

Upon secretion, effectors can modulate host processes by interacting with target host proteins. Due to its robust secretion during infection, localization to the ER, and interesting structural motifs, we decided to focus on SrfD for further investigation. To identify potential SrfD binding partners during infection, we immunoprecipitated endogenous SrfD from WT *R. parkeri*-infected host cytoplasmic lysates and performed mass spectrometry on the resulting protein complexes. As a control, we also processed lysates from uninfected host cells. In addition to SrfD itself, we found that the α and β subunits of the host Sec61 complex were highly enriched in the infected lysate pull-downs (Fig. 5a, Supplementary Data 2), suggesting that SrfD interacts with Sec61 at the ER. To verify the SrfD-Sec61 interaction, we performed the reverse pull-down and confirmed that SrfD is immunoprecipitated with Sec61β during infection (Fig. 5b). To determine if the SrfD-Sec61 interaction could be recapitulated in the absence of infection, we transiently expressed 3xFLAG-SrfD in HEK293T cells and repeated our Sec61β immunoprecipitation assays. We found that 3xFLAG-SrfD immunoprecipitated with Sec61β (Fig. 5c), demonstrating the functional relevance of our exogenous expression strategy.

The Sec61 complex forms a channel for protein translocation across the ER membrane[65], and several naturally occurring small molecules have been identified that bind and inhibit Sec61[66]. Given

that SrfD also interacts with Sec61, we tested if SrfD influences protein translocation through Sec61. We transfected 3xFLAG-SrfD into HEK293T cells stably expressing the signal peptide-containing Sec61 substrate *Gaussia* luciferase and then measured luciferase activity in culture supernatants (Supplementary Fig. 4)[67]. As a control, we treated cells with brefeldin A, which disrupts ER-Golgi trafficking and thereby blocks luciferase secretion to the cell exterior[68]. We found that luciferase secretion was unaffected in SrfD-expressing cells, suggesting that SrfD does not phenocopy the behavior of known Sec61 inhibitors.

## Multiple domains of SrfD support its interaction with Sec61 and localization to the ER

SrfD does not resemble known components of the Sec61 translocon or associated proteins, but it does harbor several putative protein-protein interaction domains. SrfD is predicted to contain two pentapeptide repeat domains (PPR1 and PPR2) and two coiled-coil motifs (CC1 and CC2), which often serve as interfaces for binding protein partners[55,56]. We hypothesized that the interaction between SrfD and Sec61 is mediated by one or more of these domains. To test this hypothesis, we immunoprecipitated Sec61β from HEK293T cells exogenously expressing one of several 3xFLAG-SrfD deletion constructs and assessed the pull-down of the constructs (Fig. 5c). We found that the

CC2 domain and predicted C-terminal transmembrane (TM) helices were mostly dispensable for the SrfD-Sec61 interaction. Within the SrfD N-terminus, however, PPR1, CC1, and PPR2 were each independently necessary for this interaction. These results suggest that the tested domains all contribute to the SrfD-Sec61 interaction, but ablation of any one of the SrfD N-terminal domains fully disrupts the interaction.

Given that SrfD localizes to the ER and interacts with Sec61, we considered two models for how SrfD localizes to the ER. In the first, the N-terminal domains alone drive localization to this compartment via their interaction with Sec61. Alternatively, the combination of the N-terminus and the predicted TM helices confers ER localization, especially because TM helices in other secreted effectors are known to promote insertion into target membranes[69,70]. To test this hypothesis, we exogenously expressed the 3xFLAG-SrfD deletion constructs in HeLa cells and used immunofluorescence microscopy to assess their localization. As expected, full-length 3xFLAG-SrfD localized with Sec61β at the ER (Fig. 5d). Interestingly, each of the domains we tested was dispensable for ER-targeting: deletion mutants that were unable to interact with Sec61β still localized to the ER, and SrfD lacking its TM domain likewise remained at the ER. These results suggest that targeting of SrfD to the ER is dependent on multiple domains and that its localization is phenotypically separable from the interaction with Sec61.

## Discussion

*Rickettsia* spp. are exquisitely adapted to their host cell niche, but the limited genetic and bioinformatic toolkit for studying these bacteria has hindered investigation of the host-pathogen interface. Here, we use cell-selective BONCAT for the first time in an obligate intracellular bacterium and greatly expand the number of experimentally validated rickettsial effectors. The Srfs we identified include *Rickettsia*-specific proteins of unknown function that are structurally diverse, variably conserved, and targeted to distinct host cell compartments. Furthermore, we rigorously validated Srf secretion into the host cell milieu through multiple orthogonal assays. Altogether, our results offer new routes to explore the unique biology of these bacterial pathogens.

The identification of Srf binding partners is an important step towards understanding their functions. For example, we found that SrfD localizes to the ER where it interacts with the host Sec61 translocon. The SrfD-Sec61 interaction was identified during infection and was recapitulated by exogenous SrfD expression in uninfected cells, providing a useful platform for structure-function analysis. Pentapeptide repeats and coiled coils are known to support protein-protein interactions, and our finding that the SrfD PPRs and CC1 are necessary for its interaction with Sec61 agrees with this point. PPRs from diverse proteins make direct contact with binding partners or otherwise serve as rigid scaffolds for other interaction domains[55]. Whether the SrfD PPRs directly interact with Sec61 or serve as spacers to position CC1 for such interaction remains to be determined. Nevertheless, we cannot exclude the possibility that SrfD interacts indirectly with Sec61 through these domains. We did not identify a candidate protein that could bridge SrfD and Sec61 from our mass spectrometry results, but future studies may reveal if the SrfD-Sec61 interaction can be reconstituted in vitro. The functional consequence of the SrfD-Sec61 interaction likewise requires continued investigation. Although SrfD did not impact the secretion of a known Sec61 substrate, it is possible that SrfD affects the translocation of other proteins in a client-selective manner. Alternatively, SrfD may influence the role played by the Sec61 translocon in other cellular processes, such as ER stress and calcium homeostasis[71,72]. SrfD may also interact with Sec61 to exert its activity on other host targets at the ER, although we note that SrfD was specifically enriched only with the Sec61 complex.

Of the Srfs we identified, only SrfA has a predicted function. SrfA is likely a functional peptidoglycan amidase in vivo, and the high abundance of the *R. conorii* SrfA homolog RC0497 during infection makes it a promising biomarker for spotted fever rickettsioses[73]. RC0497 has been observed in the periplasm of purified rickettsiae by immunogold labeling[49], despite the absence of a Sec or Tat signal peptide[74]. Our detection of SrfA in the infected host cytoplasm and phosphorylation of GSK-tagged SrfA suggest that this protein reaches its final destination outside the bacteria during infection.

For the remaining Srfs, we combined in silico predictions and localization analyses to begin characterizing these effectors. Curiously, SrfB localized to mitochondria when exogenously expressed, even though it lacks a predicted mitochondrial targeting sequence[75,76]. Once this behavior is validated for the secreted protein, mutational and biochemical studies may identify a cryptic targeting sequence within SrfB or a mitochondrial binding partner of SrfB that mediates its localization. We also showed that the localization pattern for an effector can vary depending on the source of its expression. For example, exogenous SrfC readily colocalized with the ER whereas endogenous, secreted SrfC exhibited perinuclear staining in a minority of infected cells. The localization patterns of endogenous and exogenous SrfE likewise differed. These results suggest that infection-specific cues dictate effector localization or, more simply, that exogenous expression disassociates secretion of a Srf from its delivery to a particular subcellular compartment. In contrast to SrfC and SrfE, both endogenous and exogenous SrfF exhibited similar localization to the host cytoplasm. This congruence suggests that the exogenous expression of SrfF serves as a convenient proxy for studying its secreted counterpart. As for SrfD, such localization studies could inform future efforts to identify Srf binding partners. Ultimately, biochemical characterization of Srf interactions with their targets may uncover novel mechanisms by which bacterial pathogens subvert host processes and yield new tools to probe eukaryotic cell biology.

Studying Srf evolution, expression, and impact on the rickettsial life cycle could also prove informative. Although the Srfs are unevenly distributed across the genus, the genomic regions surrounding the *srf* loci are well-conserved; this suggests that the *srf* genes and their flanking genes may face different selective pressures. The presence and diversification of these unique effectors in bacteria with notoriously streamlined genomes raise exciting questions about rickettsial evolution within the host cell niche[77]. Bioinformatic studies have traced the emergence, maintenance, and decay of genes across *Rickettsia* spp.[37,78–80], and the Srfs could serve as useful focal points for such analyses in future work. Additionally, a thorough characterization of Srf expression could clarify their roles across the genus and across host cell niches. For example, *R. conorii srfB* is highly transcribed during infection of human and tick cells[81], and the expression of *R. rickettsii srfE* and *R. typhi srfG* appears to respond to temperature shifts[82,83]. We eagerly await the generation of *srf* mutants, the study of which will provide insight into how these effectors contribute to the rickettsial lifestyle and pathogenesis. Transposon mutants for *srfB* from *R. parkeri* and *srfF* from *R. prowazekii* and *R. felis* have been isolated but remain uncharacterized[84–86]. Altogether, the results from these future studies will offer a more comprehensive view of Srf biology.

*Rickettsia* spp. harbor T4SS and T1SS machinery that may drive Srf translocation to the host cell milieu[58,87]. Our approach to identifying effectors is secretion system-agnostic, and future studies should elucidate the mechanisms by which SrfA−G are secreted during infection. Even for well-studied pathogens, however, the signal sequences for substrates of these secretion systems are not universal, and often they are genus- or species-specific[36,88–90]; thus, robust computational prediction of effectors secreted by *Rickettsia* spp. is challenging. The fact that the Srfs were not predicted by in silico tools underscores this limitation. As rickettsial secretion mutants have yet to be reported, heterologous expression or co-immunoprecipitation with components of the secretion apparatus may implicate a cognate secretion system for each Srf[13,17,18,30]. Indeed, *srfG* lies immediately upstream of a

gene encoding another DUF5410-containing protein (*cREM-2a*), which was identified as an interaction partner of the *R. typhi* T4SS component RvhD4[17]. The gene pair is predicted to have arisen via an ancient duplication event[37], and future studies may reveal if they encode bona fide T4SS effectors. Such information will ultimately help define the sequence determinants for rickettsial secretion and improve our ability to predict new effectors.

In this work, we used BONCAT to discover new *R. parkeri* effectors, but our study is by no means exhaustive. First, BONCAT does not provide truly unbiased coverage of the proteome. It has been observed that longer, Met-rich proteins are slightly overrepresented in such pull-downs[43], likely due to greater probabilities for Anl incorporation and peptide detection. Moreover, extensive replacement of amino acids with non-canonical analogs could impact protein folding, stability, secretion, and, ultimately, bacterial physiology. Second, our selective lysis strategy precludes extraction of putative effectors that localize to insoluble subcellular compartments, whose transient presence in the host cytoplasm may be insufficient for detection. Third, we labeled infected cells that had already accumulated considerable rickettsial burdens over the course of two days, a timepoint that could theoretically exclude effectors that are only secreted early during infection. Low spectral counts for the effectors we detected suggest that there is room for further optimization.

Nevertheless, we envision cell-selective BONCAT as a valuable tool for investigating rickettsial biology. For example, pulse labeling with Anl could reveal the kinetics of effector secretion across the rickettsial life cycle, as was demonstrated for Yop effector secretion during *Yersinia* infection[41]. Given that the Srfs we identified are variably conserved across the genus, effector repertoires could be compared between different *Rickettsia* species. BONCAT may also reveal that different suites of effectors are secreted during rickettsial infection of vertebrate host and arthropod vector cell niches. Additionally, arthropods harbor a multitude of microbes that can influence rickettsial biology[91–93], and in situ, strain-specific labeling could facilitate studies of *Rickettsia* spp. within the broader context of the vector microbiome.

In sum, our work demonstrates that cell-selective BONCAT can uncover novel effectors secreted by an obligate intracellular bacterial pathogen. Proteomics provide a powerful lens through which to interrogate the biology of *Rickettsia* spp. and will complement advances in genetic tool development. The identification of SrfA–G opens new avenues for exploring effector structures, diversification, and secretion by this enigmatic genus. In parallel, mapping the host cell targets of these effectors will help illuminate the host-pathogen interface and offer a handle for studying fundamental cell biological processes. Altogether, a thorough investigation of secreted effectors will enhance our understanding of rickettsial biology and pathogenesis.

## Methods

### Cell culture
A549 human lung epithelial, HeLa human cervical epithelial, HEK293T human embryonic kidney epithelial, and Vero monkey kidney epithelial cell lines were obtained from the University of California, Berkeley Cell Culture Facility (Berkeley, CA). A549, HeLa, and HEK293T cells were maintained in Dulbecco's modified Eagle's medium (DMEM; Gibco catalog number 11965118) supplemented with 10% fetal bovine serum (FBS). Vero cells were maintained in DMEM supplemented with 5% FBS. Cell lines were confirmed to be mycoplasma-negative in a MycoAlert PLUS assay (Lonza catalog number LT07-710) performed by the Koch Institute High-Throughput Sciences Facility (Cambridge, MA).

### Plasmid construction
Strains and plasmids used in this study are listed in Supplementary Table 1. pRL0128 was made by cloning *E. coli metG(M1–K548)* with

mutations *L13N*, *Y260L*, and *H301L* and codon-optimized for *R. parkeri* into pRAM18dSGA[MCS] (kindly provided by Ulrike Munderloh). To enable expression of this gene in *R. parkeri*, a 368 bp fragment upstream of the *R. parkeri metG* (MC1_RS05365) start codon and a 99 bp fragment downstream of the *R. parkeri metG* stop codon were also added. pRL0368–374 were made by cloning the *R. parkeri ompA* promoter, an N-terminal MSGRPRTTSFAESGS sequence (GSK epitope tag underlined), *srfA–G*, and the *ompA* terminator into pRAM18dSGA[MCS]. pRL0375–377 were made by cloning *E. coli* codon-optimized *srfC*, *srfD(ΔF766–N957)*, and *srfF*, respectively, into pGEX6P3 (kindly provided by Matthew Welch) to add an N-terminal GST tag. pRL0378 and pRL0379 were made by cloning *E. coli* codon-optimized *srfD(ΔF766–N957)* and *srfF*, respectively, into His-SUMO-dual strep-TEV-PGT (kindly provided by Barbara Imperiali)[94] to add an N-terminal 6xHis-SUMO-TwinStrep tag. pRL0430 was kindly provided by Supratim Dey and Karla Satchell. pRL0381 was made by replacing the Cas9 insert in HP138-puro (kindly provided by Iain Cheeseman)[95] with an MCS downstream of the anhydrotetracycline (aTc)-inducible TRE3G promoter. pRL0382, pRL0385, pRL0387, and pRL0388 were made by cloning an N-terminal MDYKDHDGDYKDHDIDYKDDDDKLIN sequence (3xFLAG epitope tag underlined) and human codon-optimized *srfA*, *srfD*, *srfF*, and *srfG*, respectively, into pRL0381. N-terminally tagged SrfB, SrfC, and SrfE expressed poorly, so pRL0383, pRL0384, and pRL0386 respectively contain a C-terminal GGSGSDYKDHDGDYKDH-DIDYKDDDDK sequence instead. FCW2IB-BiP-mNeonGreen-KDEL was generated as previously described[96]. pRL0389 was made by replacing the Lifeact-3xTagBFP insert in FCW2IB-Lifeact-3xTagBFP[12] with *Gaussia*-Dura luciferase from pCMV-Gaussia-Dura Luc (Thermo Fisher Scientific catalog number 16191). pRL0390–394 are identical to pRL0385 but *srfD* was replaced with *srfD(ΔA57–Y116)*, *srfD(ΔN126–F257)*, *srfD(ΔF305–D686)*, *srfD(ΔF766–T821)*, and *srfD(ΔK848–D890)*, respectively.

### Generation of *R. parkeri* strains
Parental *R. parkeri* strain Portsmouth (kindly provided by Chris Paddock) and all derived strains were propagated by infection and mechanical disruption of Vero cells grown in DMEM supplemented with 2% FBS at 33 °C as previously described[12]. Bacteria were clonally isolated and expanded from plaques formed after overlaying infected Vero cell monolayers with agarose as previously described[97]. When appropriate, bacteria were further purified by passage through a sterile 2 μm filter (Cytiva catalog number 6783-2520). Bacterial stocks were stored as aliquots at −80 °C to minimize variability due to freeze-thaws, and titers were determined by plaque assay[12]. Parental *R. parkeri* was transformed with plasmids by small-scale electroporation as previously described[44]. WT and MetRS* *R. parkeri* were generated by transformation with pRAM18dSGA[MCS] and pRL0128, respectively. *R. parkeri* expressing GSK-tagged BFP and RARP-2 were generated as previously described[44]. *R. parkeri* expressing GSK-tagged SrfA–G were generated by transformation with pRL0368–374. Spectinomycin (50 μg/mL) was included to select for transformants and to ensure plasmid maintenance during experiments.

### Infectious focus assays
Infectious focus assays were performed as previously described[44]. For each strain, 15 foci were imaged, and the number of infected cells and bacteria per focus was calculated.

### BONCAT microscopy validation
Confluent A549 cells (approximately $3.5 \times 10^5$ cells/cm²) were grown on 12-mm coverslips in 24-well plates and were infected with WT or MetRS* *R. parkeri* at a multiplicity of infection (MOI) of 0.001–0.004, centrifuged at $200 \times g$ for 5 min at room temperature (RT) and incubated at 33 °C. After 45 h, infected cells were treated with or without 1 mM azidonorleucine (Anl, Iris Biotech catalog number HAA1625) for

3 h, washed three times with phosphate-buffered saline (PBS), and fixed with 4% paraformaldehyde (PFA) in PBS for 10 min at RT. Fixed samples were incubated with 100 mM glycine in PBS for 10 min at RT to quench residual PFA. Samples were then washed three times with PBS, permeabilized with 0.5% Triton X-100 in PBS for 5 min at RT, and washed again with PBS. Samples were incubated with blocking buffer (2% bovine serum albumin [BSA] and 10% normal goat serum in PBS) for 30 min at RT. Primary and secondary antibodies were diluted in blocking buffer and incubated for 1 h each at RT with three 5-min PBS washes after each incubation step. The following antibodies and stains were used: mouse anti-*Rickettsia* 14-13 (kindly provided by Ted Hackstadt), goat anti-mouse conjugated to Alexa Fluor 488 (Invitrogen catalog number A-11001), and Hoechst stain (Invitrogen catalog number H3570) to detect host nuclei. To perform the click reaction, coverslips were subsequently fixed with 4% PFA in PBS for 5 min at RT, quenched with 0.1 M glycine in PBS for 10 min at RT, washed three times with PBS, incubated with lysozyme reaction buffer (0.8× PBS, 50 mM glucose, 5 mM EDTA, 0.1% Triton X-100, 5 mg/mL lysozyme [Sigma-Aldrich catalog number L6876]) for 20 min at 37 °C to permeabilize bacteria, and then washed five times with PBS. Samples were incubated with click reaction staining cocktail (50 mM sodium phosphate buffer [pH 7.4], 4 mM copper (II) sulfate [Sigma-Aldrich catalog number 209198], 20 mM tris-(3-hydroxypropyltriazolylmethyl)amine [THPTA, Sigma-Aldrich catalog number 762342], 5 µM AZDye 647 alkyne [Click Chemistry Tools catalog number 1301], 10 mM sodium ascorbate [Sigma-Aldrich catalog number A4034]) for 30 min at RT and washed five times with PBS. Coverslips were mounted using ProLong Gold Antifade mountant (Invitrogen catalog number P36934) and images were acquired using a 100× UPlanSApo (1.35 NA) objective on an Olympus IXplore Spin microscope system. Image analysis was performed with ImageJ.

### BONCAT western blot validation

Confluent A549 cells (approximately $3.5 \times 10^5$ cells/cm$^2$) were grown in 6-well plates and were infected with WT or MetRS* *R. parkeri* at an MOI of 0.006–0.02, centrifuged at $200 \times g$ for 5 min at RT, and incubated at 33 °C. After 45 h, infected cells were treated with or without 1 mM Anl for 3 h, washed with PBS, lifted with trypsin-EDTA, and centrifuged at $2400 \times g$ for 5 min at RT. Infected cell pellets were washed three times with PBS, resuspended in selective lysis buffer (50 mM HEPES [pH 7.9], 150 mM NaCl, 10% glycerol, 1% IGEPAL) supplemented with protease inhibitors (Sigma-Aldrich catalog number P1860), incubated on ice for 20 min, and centrifuged at $11,300 \times g$ for 10 min at 4 °C. The resulting supernatants were passed through a 0.22-µm cellulose acetate filter (Thermo Fisher Scientific catalog number F2517-1) by centrifugation at $6700 \times g$ for 10 min at 4 °C. The resulting pellets were resuspended in total lysis buffer (50 mM HEPES [pH 7.9], 150 mM NaCl, 10% glycerol, 2% sodium dodecyl sulfate [SDS]) supplemented with 2 mM MgCl$_2$ and 0.1 units/µL Benzonase (Sigma catalog number E1014), incubated for 5 min at 37 °C, and clarified by centrifugation at 21,100 x g for 1 min at RT. Lysate protein content was determined by bicinchoninic acid assay (Thermo Fisher Scientific catalog number 23227) and 45 µg cytoplasmic lysate was used as input for a click reaction at 1 mg/mL protein. An equivalent volume of pellet lysate was used as input. Lysates were incubated with click reaction biotin-alkyne cocktail (50 mM sodium phosphate buffer [pH 7.4], 2 mM copper (II) sulfate, 10 mM THPTA, 40 µM biotin-alkyne [Click Chemistry Tools catalog number 1266], 5 mM aminoguanidine [Sigma-Aldrich catalog number 396494], 20 mM sodium ascorbate) for 90 min at RT and proteins were precipitated with chloroform/methanol. Clicked protein precipitates were boiled in loading buffer (50 mM Tris-HCl [pH 6.8], 2% SDS, 10% glycerol, 0.1% bromophenol blue, 5% β-mercaptoethanol) and detected by Western blotting using StrepTactin-HRP (Bio-Rad catalog number 1610381). To prevent signal saturation, only 10% of the pellet precipitate was loaded.

### BONCAT pull-downs

Confluent A549 cells (approximately $3.5 \times 10^5$ cells/cm$^2$) were grown in 10 cm$^2$ dishes and were infected with MetRS* *R. parkeri* at a MOI of 0.3, gently rocked at 37 °C for 50 min, and incubated at 33 °C. After 48 h, infected cells were treated with or without 1 mM Anl ($n = 2$ dishes per condition) for 5 h, washed with PBS, lifted with trypsin-EDTA, and centrifuged at $2400 \times g$ for 5 min at RT. Cytoplasmic lysates were harvested as described in the "BONCAT western blot validation" section, SDS was added to 4.2 mg lysate input to a final concentration of 1.7%, and the mixture was heated at 70 °C for 10 min. Denatured lysates were diluted to 1 mg/mL protein and 0.4% SDS with click reaction biotin-alkyne cocktail, incubated for 90 min at RT, and precipitated with 20% trichloroacetic acid. Clicked protein precipitates were washed with acetone, resuspended to 1.4 mg/mL protein in 1% SDS in PBS, and carryover acid was neutralized with 118 mM Tris-HCl (pH 8). To stabilize streptavidin tetramers during pull-down and washes, cross-linked streptavidin resin was prepared following a previously described resin cross-linking strategy[98]. Briefly, streptavidin resin (Cytiva catalog number 17511301) was incubated with cross-linking buffer (20 mM sodium phosphate [pH 8], 150 mM NaCl) supplemented with 1.2 mM bis(sulfosuccinimidyl)suberate (BS3, Thermo Fisher Scientific catalog number A39266) for 30 min at RT. Unreacted BS3 was quenched with 40 mM Tris-HCl (pH 8) for 15 min at RT and the cross-linked streptavidin resin was washed twice with resin wash buffer (25 mM Tris-HCl [pH 7.4], 137 mM NaCl, 0.1% Tween 20) and once with PBS. Clicked protein suspensions were incubated with 200 µL cross-linked streptavidin resin for 2 h at RT, washed four times with 1% SDS in PBS, once with 6 M urea in 250 mM ammonium bicarbonate, once with 1 M NaCl, twice with 0.1% SDS in PBS, and five times with PBS. Resin-bound proteins were submitted to the Whitehead Institute Proteomics Core Facility (Cambridge, MA) for sample workup and mass spectrometry analysis.

### GSK secretion assays

GSK secretion assays were performed as previously described[44]. Briefly, confluent Vero cells (approximately $4 \times 10^5$ cells/cm$^2$) were grown in 24-well plates, infected with the indicated strains at a MOI of 0.02–0.08, centrifuged at $200 \times g$ for 5 min at RT, and incubated at 33 °C. Vero cells were chosen for their routine use in propagating rickettsiae and performing rickettsial GSK secretion assays. After 72 h, infected cells were washed with ice-cold serum-free DMEM, directly lysed in loading buffer, and boiled. Lysates were analyzed by Western blotting using rabbit anti-GSK-3β-Tag (Cell Signaling Technology catalog number 9325S) and rabbit anti-phospho-GSK-3β (Cell Signaling Technology catalog number 9336S).

### Srf protein purification

GST-tagged constructs were expressed in *E. coli* BL21 by overnight induction with 0.3 mM IPTG at 18 °C. Pelleted cells were resuspended in protein lysis buffer (50 mM HEPES [pH 8.0], 150 mM NaCl, 0.1% Triton X-100, 1 mM PMSF, 6 units/mL Benzonase, 6 mM MgCl$_2$) supplemented with protease inhibitor tablets (Sigma-Aldrich catalog number 11836153001), lysed using a LM20 Microfluidizer (Microfluidizer) at 18,000 PSI for three passes, and clarified by centrifugation at $40,000 \times g$ for 1 h at 4 °C. Proteins were purified using glutathione sepharose resin (Cytiva catalog number 17075601), eluted by step gradient (50 mM HEPES [pH 8], 200 mM NaCl, 1–10 mM reduced glutathione), and concentrated using Amicon Ultra concentrators (Sigma-Aldrich). 6xHis-SUMO-TwinStrep-tagged constructs were expressed in *E. coli* BL21(DE3) and harvested as described for the GST-tagged proteins, purified using nickel sepharose resin (Cytiva catalog number 17531802), eluted by step gradient (50 mM HEPES [pH 8], 200 mM NaCl, 100–500 mM imidazole), and cleaved with ULP1 protease (kindly provided by Barbara Imperiali) while dialyzing overnight at 4 °C (into 50 mM HEPES [pH 8], 200 mM NaCl). TwinStrep-tagged proteins were

further purified using nickel sepharose resin followed by size-exclusion chromatography using a HiLoad 16/600 Superdex 200 pg column (Cytiva catalog number 28989335) and then concentrated.

A 6xHis-MBP-TEV-tagged SrfE(277–403) construct was expressed in *E. coli* BL21(DE3) by overnight induction with 0.2 mM IPTG at RT. Pelleted cells were resuspended in SrfE lysis buffer (25 mM Tris [pH 8.3], 500 mM NaCl, 0.1% IGEPAL, 1 mM TCEP, 10% glycerol, 6 units/mL Benzonase, 6 mM $MgCl_2$) supplemented with protease inhibitor tablets and then lysed and clarified as described above. Proteins were purified using a prepacked nickel sepharose resin column (Cytiva catalog number 17525501) and eluted by linear gradient (25 mM Tris [pH 8.3], 500 mM NaCl, 0.5 mM TCEP, 10% glycerol, 500 mM imidazole) onto amylose resin (New England Biolabs catalog number E8022S). Proteins were then eluted (25 mM Tris [pH 8.3], 500 mM NaCl, 0.5 mM TCEP, 10% glycerol, 20 mM imidazole, 116 mM maltose) and cleaved with TEV protease (kindly provided by Seychelle Vos) while dialyzing overnight at 4 °C (into 25 mM Tris [pH 8.3], 250 mM NaCl, 0.5 mM TCEP, 10% glycerol). Cleaved proteins were further purified using a prepacked nickel sepharose resin column followed by size-exclusion chromatography and then concentrated and dialyzed overnight at 4 °C (into 50 mM HEPES [pH 8], 200 mM NaCl). Purified untagged SrfE(277–403) was also kindly provided by Supratim Dey and Karla Satchell.

## Srf antibody purification

GST-tagged SrfC, SrfD, and SrfF and untagged SrfE were used for immunization by Labcorp (Denver, PA) according to their standard 77-day rabbit polyclonal antibody protocol. To affinity purify anti-Srf antibodies, NHS-activated sepharose resin (1 mL, Cytiva catalog number 17090601) was activated with 1 mM HCl, drained, and incubated with TwinStrep-tagged proteins or untagged SrfE (1.4 mg) for 1 h at RT. The resin was washed twice with alternating ethanolamine (500 mM ethanolamine [pH 8.3], 500 mM NaCl) and acetate (100 mM sodium acetate [pH 4.5], 500 mM NaCl) buffers and then equilibrated (with 20 mM Tris [pH 7.5] first with and then without 500 mM NaCl) before incubating with 2 mL filtered (0.22 μm) SrfD, SrfE, or SrfF antisera for 1 h at RT. The resin was washed (20 mM Tris [pH 7.5] first without and then with 500 mM NaCl), and antibodies were eluted with 100 mM glycine (pH 2.8), neutralized with 65 mM Tris-HCl (pH 8.8), dialyzed overnight at 4 °C (into 50 mM HEPES [pH 8], 150 mM NaCl, 10% glycerol), and concentrated. For retrieval of anti-SrfC antibodies, filtered SrfC antisera were purified by sequential incubation with GST tag and GST-tagged SrfC conjugated separately to NHS-activated sepharose resin. Antibodies were validated by Western blotting using purified *R. parkeri*, uninfected A549 cell lysates, and purified recombinant Srfs.

## Secreted Srf immunoblotting

Confluent A549 cells (approximately $3.5 \times 10^5$ cells/cm²) were grown in 10 cm² dishes and were infected with parental *R. parkeri* at a MOI of 0.2–0.5, gently rocked at 37 °C for 50 min, and incubated at 33 °C. After 48 h, infected cells were washed with PBS, lifted with trypsin-EDTA, and centrifuged at $2400 \times g$ for 5 min at RT. Cytoplasmic and pellet lysates were harvested as described in the "BONCAT western blot validation" section, boiled in loading buffer, and analyzed by Western blotting using Srf antisera and mouse anti-RpoA (BioLegend catalog number 663104).

## Secreted Srf immunofluorescence assays

Confluent A549 cells (approximately $3.5 \times 10^5$ cells/cm²) were grown on 12-mm coverslips in 24-well plates and were infected with WT *R. parkeri* at a MOI of 0.08 or 0.15, centrifuged at $200 \times g$ for 5 min at RT, and incubated at 33 °C for 47 h until fixation with 4% PFA in PBS for 1 h at RT. Fixed samples were processed as described in the "BONCAT microscopy validation" section, except primary antibodies were incubated for 3 h at 37 °C. The following antibodies and stains were used:

purified rabbit anti-Srf antibodies, goat anti-rabbit conjugated to Alexa Fluor 647 (Invitrogen catalog number A-21245), and Hoechst stain to detect host nuclei.

## Exogenous Srf immunofluorescence assays

HeLa cells ($4 \times 10^4$ cells/cm²) were plated on 12-mm coverslips in 24-well plates and were transfected the next day with 500 ng DNA using Lipofectamine 3000 (Thermo Fisher Scientific catalog number L3000001) following the manufacturer's instructions. HeLa cells were chosen to study exogenous Srf localization patterns over A549s due to their superior transfection efficiency. The following day, the media was replaced and supplemented with 1 μg/mL aTc (Clontech catalog number 631310). After 24 h induction, cells were fixed with 4% PFA in PBS for 10 min at RT. Fixed samples were quenched, washed, and permeabilized and then incubated with blocking buffer (2% BSA in PBS) for 30 min at RT. Primary and secondary antibodies were diluted in blocking buffer and incubated for 1 h each at RT with three 5-min PBS washes after each incubation step. The following antibodies and stains were used: mouse anti-FLAG (Sigma-Aldrich catalog number F1804), goat anti-mouse conjugated to Alexa Fluor 488 or to Alexa Fluor 647 (Invitrogen catalog number A-21235), rabbit anti-AIF (Cell Signaling Technology catalog number 5318S), goat anti-rabbit conjugated to Alexa Fluor 488 (Invitrogen catalog number A-11008), and Hoechst stain to detect nuclei. To assess colocalization of 3xFLAG-SrfC or 3xFLAG-SrfD with ER-targeted mNeonGreen, the same procedure was followed except 250 ng each of pRL0384 or pRL0385 and FCW2IB-BiP-mNeonGreen-KDEL were co-transfected and the cells were fixed for 1 h.

## Secreted SrfD immunoprecipitation assays

Confluent A549 cells (approximately $3.5 \times 10^5$ cells/cm²) were grown in triplicate in 10 cm² dishes and were infected with WT *R. parkeri* at a MOI of 0.3, gently rocked at 37 °C for 50 min, and incubated at 33 °C. Triplicate dishes were infected with bacteria in brain heart infusion media (BHI) or mock-infected with BHI as uninfected controls. After 45 h, cells were washed with ice-cold PBS, scraped into selective lysis buffer supplemented with protease inhibitors and 1 mM EDTA, incubated on ice for 20 min, and centrifuged at $11,300 \times g$ for 10 min at 4 °C. The resulting supernatants were filtered as described in the "BONCAT western blot validation" section, pre-cleared with Protein A magnetic resin (Thermo Fisher Scientific catalog number 88846) for 30 min at 4 °C, and incubated with 15 μg/mL purified rabbit anti-SrfD overnight at 4 °C. Immune complexes were precipitated with Protein A magnetic resin for 1 h at 4 °C, washed four times with supplemented selective lysis buffer, eluted by incubation with 100 mM glycine (pH 2.8) for 20 min at RT, and neutralized with 115 mM Tris-HCl (pH 8.5). The neutralized eluates were submitted to the Koch Institute Biopolymers & Proteomics Core Facility for sample workup and mass spectrometry analysis.

## Mass spectrometry

For identification of secreted effectors from BONCAT, resin-bound proteins were denatured, reduced, alkylated, and digested with trypsin/Lys-C overnight at 37 °C. The resulting peptides were purified using styrene-divinylbenzene reverse-phase sulfonate StageTips as previously described[99]. LC-MS/MS data were acquired using a Vanquish Neo nanoLC system coupled with an Orbitrap Exploris mass spectrometer, a FAIMS Pro interface, and an EASY-Spray ESI source (Thermo Fisher Scientific). Peptide separation was carried out using an Acclaim PepMap trap column (75 μm × 2 cm; Thermo Fisher Scientific) and an EASY-Spray ES902 column (75 μm × 250 mm, 100 Å; Thermo Fisher Scientific) using standard reverse-phase gradients. Data analysis was performed using PEAKS Studio 10.6 software (Bioinformatics Solutions) and analyzed as previously described[100]. RefSeq entries for *R. parkeri* strain Portsmouth (taxonomy ID 1105108) were downloaded from NCBI. Variable modifications for Anl and biotin-Anl were included. Peptide identifications were accepted with a false discovery rate

of ≤1% and a significance threshold of 20 (−10log$_{10}$P). Protein identifications were accepted with two unique peptides. Proteins that were present in both replicates of the Anl-treated infection lysate pull-down were called hits.

For identification of proteins in the secreted SrfD immunoprecipitation eluates, peptides were prepared using S-Trap micro spin columns (ProtiFi) following the manufacturer's instructions, except 10 mM DTT was used instead of TCEP, samples were reduced for 10 min at 95 °C, 20 mM iodoacetamide was used instead of MMTS, and samples were alkylated for 30 min at RT. LC-MS/MS data were acquired using an UltiMate 3000 HPLC system coupled with an Orbitrap Exploris mass spectrometer (Thermo Fisher Scientific). Peptide separation was carried out using an Acclaim PepMap RSLC C18 column (75 μm × 50 cm; Thermo Fisher Scientific) using standard reverse-phase gradients. Data analysis was performed using Sequest HT in Proteome Discoverer (Thermo Fisher Scientific) against human (UniProt) and *R. parkeri* (RefSeq) databases with common contaminants removed. Protein identifications were accepted with two unique peptides, and normalized intensities from the top three precursors were computed with Scaffold (Proteome Software). These results were filtered to require a non-zero value for at least two of the three replicates in at least one condition. Zero values were then imputed to the minimum intensity within each sample. Mean fold-changes and Benjamini−Hochberg adjusted *p*-values were computed for log-transformed intensities between infected and uninfected conditions.

## Sec61 immunoprecipitation assays

For immunoprecipitation of Sec61 during infection, confluent A549 cells (approximately $3.5 \times 10^5$ cells/cm$^2$) were grown in 10 cm$^2$ dishes and were infected with WT *R. parkeri* at a MOI of 0.25, gently rocked at 37 °C for 50 min, and incubated at 33 °C. After 53 h, lysates were harvested, filtered, and pre-cleared as described in the "Secreted SrfD immunoprecipitation assays" section, and incubated with 0.18 μg/mL rabbit anti-Sec61β (Cell Signaling Technology catalog number 14648S) overnight at 4 °C. Immune complexes were precipitated and washed, eluted by boiling in loading buffer and detected by Western blotting using purified rabbit anti-SrfD and rabbit anti-Sec61β. For immunoprecipitation of Sec61 following SrfD transfection, HEK293T cells ($5 \times 10^4$ cells/cm$^2$) were grown in 6-well plates and transfected the next day with 2.5 μg DNA with TransIT-LT1 (Mirus Bio catalog number MIR-2304) following the manufacturer's instructions. HEK293T cells were chosen to study exogenous SrfD over A549s due to their superior transfection efficiency and routine use in co-immunoprecipitation experiments. To ensure comparable expression levels of the SrfD constructs, 2 μg pRL0385, 2 μg pRL0390, 2 μg pRL0391, 2 μg pRL0392, 2.5 μg pRL0393, and 1.5 μg pRL0394 were brought up to 2.5 μg total DNA with pRL0381. The following day, the media was replaced and supplemented with 200 ng/mL aTc. After 24 h induction, lysates were harvested (without filtering), pre-cleared, and incubated with rabbit anti-Sec61β. Immune complexes were precipitated, washed, eluted, and detected by Western blotting using mouse anti-FLAG and rabbit anti-Sec61β.

## Luciferase secretion assays

HEK293T cells stably expressing *Gaussia* luciferase were generated with pRL0389 by lentiviral transduction as previously described[12], except 300 μL filtered viral supernatant was used and selection was performed with 5 μg/mL blasticidin. Cells ($5 \times 10^4$ cells/cm$^2$) were grown in triplicate in 24-well plates pre-coated with 6 μg fibronectin (Sigma-Aldrich catalog number FC010) and transfected the next day with 500 ng pRL0381 or pRL0385 with TransIT-LT1. The following day, the media was first replaced and supplemented with 200 ng/mL aTc to prime the expression of SrfD. After 8 h, the media was replaced and supplemented with 200 ng/mL aTc and either DMSO or 6 μg/mL brefeldin A (Sigma-Aldrich catalog number B6542). After an additional

16 h, culture supernatants were assayed for luciferase activity on a Varioskan plate reader (Thermo Fisher Scientific) using the Pierce Gaussia Luciferase Glow Assay Kit (Thermo Fisher Scientific catalog number 16161) following the manufacturer's instructions.

## Bioinformatic analyses

Protein-protein BLAST[101] was used to detect putative homologs of *R. parkeri* str. Portsmouth SrfA−G across the *Rickettsia* genus (taxonomy ID 780). E-values were computed for the following species selected for display using the default BLOSUM62 scoring matrix, and hits were curated by reciprocal BLAST against *R. parkeri* Srfs: *R. conorii* str. Malish 7 (taxonomy ID 272944), *R. rickettsii* str. Sheila Smith (taxonomy ID 392021), *R. peacockii* str. Rustic (taxonomy ID 562019), *R. montanensis* str. OSU 85-930 (taxonomy ID 1105114), *Rickettsia* endosymbiont of *Ixodes pacificus* (taxonomy ID 1133329), *R. monacensis* str. IrR/Munich (taxonomy ID 1269334), *R. buchneri* str. ISO7 (taxonomy ID 1462938), *R. felis* str. URRWXCal2 (taxonomy ID 315456), *R. akari* str. Hartford (taxonomy ID 293614), *R. prowazekii* str. Breinl (taxonomy ID 1290428), *R. typhi* str. Wilmington (taxonomy ID 257363), *R. helvetica* str. C9P9 (taxonomy ID 1144888), *R. canadensis* str. McKiel (taxonomy ID 293613), *R. bellii* str. RML369-C (taxonomy ID 336407), *Rickettsia* endosymbiont of *Bemisia tabaci* MEAM1 (taxonomy ID 1182263), and *Rickettsia* endosymbiont of *Pyrocoelia pectoralis* (taxonomy ID 2866165). Protein-protein BLAST was also used to detect putative Srf homologs in organisms excluding those of the *Rickettsia* genus. *R. parkeri* Srf structures were predicted with ColabFold[50] and searched against the Alpha-Fold, PDB, and GMGCL databases using FoldSeek[51] in 3Di/AA mode with an E-value cutoff of 0.001. HHpred[52] with an E-value cutoff of 0.001 was used to search Srf sequences against the PDB and Pfam databases. Phyre2[53] with a 95% confidence cutoff was also used for Srf homolog prediction. Putative secondary structure features were identified using the MPI Bioinformatics Toolkit[52]. The *R. parkeri* proteome was searched for type IV effectors using OPT4e[36] and S4TE[59]. Conservation of synteny for the *srf* gene neighborhoods was evaluated using the Compare Region Viewer tool from the Bacterial and Viral Bioinformatics Resource Center[102]. Searches were anchored by flanking genes to maximize alignment between genomes that are not predicted to encode a given Srf homolog. Gene neighborhoods from the following species were selected for display: *R. parkeri* str. Portsmouth, *R. rickettsii* str. Sheila Smith, *Rickettsia* endosymbiont of *Ixodes pacificus*, *R. felis* str. URRWX-Cal2, *R. typhi* str. Wilmington, *R. canadensis* str. McKiel, and *R. bellii* str. RML369-C. Gene annotations were provided by the Bacterial and Viral Bioinformatics Resource Center or manual curation. SrfA was searched for Sec and Tat signal peptides using SignalP[74]. SrfB was searched for a mitochondrial targeting sequence using TargetP[75] and MitoFates[76].

## Statistical analyses

Statistical analysis was performed using Prism 9 (GraphPad Software). Graphical representations, statistical parameters, and significance are reported in the figure legends.

## Reporting summary

Further information on research design is available in the Nature Portfolio Reporting Summary linked to this article.

## Data availability

The protein mass spectrometry data generated in this study have been deposited in the public proteomics repository MassIVE (https://massive.ucsd.edu) under accession codes MSV000093380 and MSV000093381. Source data are provided with this paper.

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

## Acknowledgements
We are grateful to Michael Laub and Brandon Sit for their critical review of the manuscript. We thank Ulrike Munderloh, Matthew Welch, Barbara Imperiali, Iain Cheeseman, Chris Paddock, and Ted Hackstadt for reagents and Roberto Vazquez Nunez and Seychelle Vos for assistance with protein purification. Additionally, we thank Supratim Dey and Karla Satchell at the Northwestern University Feinberg School of Medicine and the Center for Structural Biology of Infectious Diseases for providing purified recombinant SrfE ahead of publication. We also thank Fabian Schulte at the Whitehead Institute Proteomics Core Facility and Richard Schiavoni at the Koch Institute Biopolymers & Proteomics Core Facility for experimental support. Work in the Lamason laboratory is supported in part by the National Institutes of Health (R01 AI155489) and by the Office of the Assistant Secretary of Defense for Health Affairs through the Tick-Borne Disease Research Program (TB200032). Opinions, interpretations, conclusions, and recommendations are those of the authors and are not necessarily endorsed by the Department of Defense. This work was also supported by the NIH Institutional Training Grants for A. G. S. and H. K. M. (T32 GM007287 and GM136540) and by the National Science Foundation Graduate Research Fellowship to H. K. M. (2141064).

## Author contributions
A.G.S.: Conceptualization, Methodology, Investigation, Formal Analysis, Validation, Visualization, Writing - Original Draft, Writing - Review & Editing. H.K.M.: Investigation, Validation, Writing - Review & Editing. A. F. M.: Validation. R.L.L.: Conceptualization, Writing - Original Draft, Writing - Review & Editing, Supervision, Funding Acquisition.

## Competing interests
The authors declare no competing interests.
