## [Peer Review File · Nature Communications]

Cell-selective proteomics reveal novel effectors secreted by an obligate intracellular bacterial pathogenREVIEWER COMMENTS

Reviewer #1 (Remarks to the Author):

The manuscript titled "Cell-selective proteomics reveal novel effectors secreted by an obligate 1 intracellular bacterial pathogen" by Sanderlin et al. uses a novel approach to Rickettsiology (BONCAT) for identifying effectors that are secreted by *Rickettsia parkeri* during host cell infection. BONCAT (bioorthogonal non-canonical amino acid tagging) utilizes the need for rickettsia to uptake amino acids from the host environment by introducing a tagged amino acids that are derivatives of expressed and translocated effectors. The approach captured validated and novel effectors, with further experimentation on several novel effectors showing they localize to the host cytoplasm, mitochondria, and ER. One novel secreted rickettsial factor, SrfD, was further determined to interact with the host Sec61 translocon.

Overall, this is an excellent report that will not only be hugely impactful to Rickettsiology, but also inspire pathogenesis work on fastidious microbes that are difficult to study. The approach is clever (yet practical) and will certainly be utilized by the field for related microbes. The identification of novel effectors with experimental evidence is a big contribution to the field. This is a study from a younger PI who is steadily making significant contributions to the field of Rickettsiology, with this specific achievement unearthing insight on how rickettsiae exploit host cells to facilitate invasion, replication and intercellular spread. Clearly there is some connection with *Rickettsia* infection and targeting the host secretory pathway at the ER. The PI also has a paper on BioRx demonstrating that *R. parkeri* interacts with the ER envelope, which complements the findings here for interactions between SrfD and the the host Sec61 translocon.

I find several shortcomings with this otherwise important and well-conducted study. First, the literature review of rickettsial protein secretion is very limited. The author cites two important works (an earlier secretome review by Gillespie and colleagues) and an unpublished work by the same group (Verhoeve et al.) that contain a lot of findings and literature that should be better captured in this report. Some specific details on this oversight are mentioned below. Second, the phylogenomics analysis is very limited relative to the dozens of diverse *Rickettsia* and *Tisiphia* genomes now available. This makes the paper very *R. parkeri*-centric, with minimal comparative power and overall applicability to the genus *Rickettsia* as a whole.

Specific points raised below.

Naming novel effectors SrfS may pose a problem down the road. A nomenclature already exists for rvh T4SS effectors (REMs) and some of the SrfS could be secreted by the rvh T4SS, so they would also qualify as REMS. Since this study characterized one species, *R. parkeri*, it may be more prudent to name the seven SrfS something more specific to this species, as has been done for *Anaplasma* T4SS effectors when annotations on protein function cannot be made. This is just a consideration. "Secreted" also really applies to surface localized proteins.

The Abstract is nicely written. However, providing numbers for known and novel effectors would be helpful. For instance, how many effectors identified by BONCAT have some other experimental support for being secreted (by any *Rickettsia* species)? Of the novel effectors, have any been captured in screens from other papers (those using biochemical approaches to identify secreted proteins)? As written, it seems like there are many more effectors than there actually are, but without listing the exact contribution to the effector landscape, the readers will really have no idea what gains this work has accomplished.

Line 53-60: This really falls short of a synopsis for the protein secretion research on rickettsiae. The authors cite a recent paper (Verhoeve et al.) that lists many more bona fide and candidate T4SS effectors. Another paper from the same group that the authors also cite (Gillespie et al.) covers all secretory proteins in greater detail, including the work from Azad that analyzed the SEC signals of many Sec-translocated proteins. Still, there are several works that use tags or whole protein antibodies to show secretion during host cell infection. Some of these studies have used B2H and/or Co-IP/pull-down to show effector interactions with either host molecules or secretion machine components. All this literature cannot go undiscussed. As written, this section is a highly

inaccurate depiction of the status of the field.

Line 75-82: Again, not stating clearly what the overall contribution of the work is (numbers of newly identified effectors) makes the significance hard to realize.

Demonstrating MetRS* expression does not impede cell-to-cell spread or bacterial growth, or seemingly overall expression or bacterial fitness, is appreciated.

The MASS SPEC results seem very clean. Is it standard to not include all peptides from the analysis? Check with the journal requirements. Also, some of the proteins listed as having minimal hits are known to be secreted (RickA and Sca5). Sca5 in particular is probably present more than any other protein on the surface of rickettsial cells and is also processed. It is also highly expressed, so AnI incorporation should be very high. RickA is well characterized by the Welch group and others and is active during *R. parkeri* infection. Why are these proteins not labeled with AnI like the other effectors?

FIGURE 1: Nicely done and convincing that BONCAT was successful. For panel C, the bacteria should be further described as purified or partially purified, no?

FIGURE 2: the cladogram depicting the phylogeny of the major Rickettsia groups does not agree with current robust genome-based phylogeny estimations. The TRG and TG are monophyletic by the works of Gillespie, Ettema, and Hurst and their teams, and their phylogeny estimations are very robust. This is also a poor representation of the actual Rickettsia diversity, but it may not be the goal of the authors to be thoroughly robust yet just list a couple examples. AG, is an old term and does not include any natural monophyletic group. BG refers to the bellii group, which now has nearly a dozen genomes. But *R. canadensis* is not a member of AG. The authors could just remove the cladogram and AG and the graphic would be fine. Or redraw the tree and leave out AG.

Of the seven Srf proteins, how many have other experimental evidence for expression, secretion, subcellular localization, mutagenesis, etc. from studies in the literature? What about their homologs in other Rickettsia species? How about the best BlastP results against non-Rickettsia organisms? In current form, there is not much information provided from the bioinformatics analyses that were conducted.

Also, entering the locus tags into NCBI does not return any data. The authors should use the current GenBank accession numbers so that the reader can access the sequences without problems. They state that the Portsmouth strain was used but the accession numbers for the Srf proteins are not provided. The WP_ numbers in Supplemental Table 1 can be used to trace down the proper locus tags or individual accession numbers. For instance, MC1_RS02555 is the locus tag MC1_02805 and accession number AFC74691.1. These numbers would be better in Figure 2 than those listed. Unless I am missing something.

Line 145-146: SrfG has been called cREM-2a in the cited preprint by Verhoeve et al.

Line 148: Is there any information from genome synteny or gene neighborhood across Rickettsia species that can link some of the Srf proteins to other proteins that may be involved in host interactions? Are these gene neighborhoods conserved across species or variable (e.g., is there any evidence for recombination hotspots, pseudogene graveyards, etc.)? Supplemental Figure 2 is not very helpful for understanding gene neighborhood or relative conservation of Srf neighborhoods across Rickettsia genomes. The BVR-BRC compare region viewer tool could help here.

Specific comments on the Srf proteins:

SrfA-D are absent in TG rickettsiae and overall seem to show a patchwork distribution across the remaining Rickettsia genomes. It might be nice to expand on this regarding pseudogenization in Rickettsia genomes and how lineage specific factors "come-and-go" via LGT and gene decay.

For SrfD, the NCBI CDD returns a match to "type IVB secretion system protein DotG/IcmE". This is a scaffold protein that utilizes the pentapeptide motif to interact with other scaffold components

within the I-T4SSs. This may indicate a function for SrfD in building a scaffold for translocation across the ER membrane. This could be worth mentioning in the discussions on SrfD-Sec61 interactions.

SrfA is a predicted PGN hydrolase and the *R. conorii* homolog that has that activity; why, then, is SrfA secreted? Does it not act on *R. parkeri* PGN but PGN of congener microbes (interbacterial antagonism)? This protein likely doesn't reach the periplasm (although the cited paper says otherwise), yet probably functions equivalent to AmpD, which degrades PGN subunits in the cytoplasm after they are imported via AmpG during murein turnover. Regardless, what would its target be in the eukaryotic cytosol? Moonlighting?

Line 154: Indeed! Major point of the paper here.

Line 169: Is there any information for SrfB and SrfE expression/secretion from the literature?

Line 174: Why were antibodies not raised against SrfA and SrfG? Since they were expressed, why not determine their subcellular localization?

Line 187: It might help to distinguish the host cell cytoplasm from the bacterial cytoplasm by calling the former "cytosol" and the latter "cytoplasm". It can be confusing otherwise, or just use the proper adjective when using "cytoplasm".

Line 201: Along these lines, should it be stated that ectopic expression sans infection will dissociate the secretion pathways for Srfs, which may be important for the accurate delivery to their intended host targets. So, the observed subcellular localization patterns might be different if the same experiments were done during infection.

Line 212: The experimental evidence for the DrfD-Sec61 interaction is robust.

The Discussion is nicely written.

Line 329: The most current reference for this system is PMID: 27307105.

Line 332-334: The signals for most T4SS effectors have been shown to be genus- or species-specific, and they are typically confirmed by experimental assays. Just because programs exist to predict non-SEC secreted effectors does not mean any of them are decent or trained on anything conserved or universal (unlike the SEC and TAT signals that are highly conserved and amenable to bioinformatics tools). The way this is written implies that other organisms have in silico tools that work for predicting non-SEC secreted proteins. This is simply not true. And the algorithms do not perform well on other species when they do show some promise for one particular species (like a workhorse that has a lot of effectors experimentally characterized, so the models and algorithms work in that particular system).

Line 336-342: the named SrfG effector (AFC74686) contains the DUF5410 domain that is unique to *Rickettsia* species. Verhoeve et al. (bioRxiv. 2023 Feb 27:2023.02.26.530123. doi: 10.1101/2023.02.26.530123. Preprint) recently analyzed these proteins and found them to belong to a multi-gene family (Figure 7b in that report). The authors cite this work, but the effectors were named cREM-2 with letters denoting their distinct groups based on phylogeny estimation and domain analyses. It would be helpful for the community and literature if the "cREM2" nomenclature were used or at least referenced. This could also be an opportunity for the authors to take a stab at mentioning the "REM" and "cREM" annotations from Verhoeve et al. and how their "Srf" annotations fit in. It would be great to acknowledge the work from both groups and call for consensus. Also, the fact that cREM-2a of *Rickettsia typhi* (locus tag RT0352) bound RvhD4 in a pull-down assay with *R. typhi* lysate, yet the adjacent gene encoding the cREM-2b protein (locus tag RT0351) accentuates that all of these experimental approaches are not "all-or-none". The pull-down from the Azad lab papers that captured REMs and cREMs interacting with RvhD4 missed one of the effectors captured by Sanderlin et al., RARP-2, which has not only been shown to bind RvhD4 in one-to-one Co-IPs and B2H assays, but that interaction is abolished when the C-terminal RARP-2 tail is removed (Lehman et al., PMID: 29946049). It would be good for the authors to

acknowledge that all of these experimental methods have limitations, and that results across different labs using different species and strains may not corroborate one another, but any experimental data nonetheless is important to call out.

Greater depth with phylogenomics could help discern those Srfs present in bona fide non-pathogens like *R. peacockii*, REIP, and *R. buchneri*. Based on the current presentation, it isn't clear if any of the Srfs can possibly define interactions with vertebrate cells and not both vertebrate and arthropod cells (the non-pathogens wouldn't need Srfs if they are only required for vertebrate cell takeover).

Methods seem fine. Remaining figures and supplements are fine.

Reviewer #2 (Remarks to the Author):

In this manuscript, Sanderlin et al. successfully demonstrated the application of BONCAT for *R. parkeri*, a tick-transmitted obligate intracellular pathogen, and identified novel molecules deposited in various subcellular compartments during *R. parkeri* replication in mammalian host cells. This is a huge step forward for the field where traditional genetic and bioinformatic tools demonstrate low efficiencies in identifying novel factors and characterizing their molecular functions impacting host-pathogen interactions. The manuscript is well-written, with supportive data and detailed descriptions of experimental methods. Below are a few questions and comments that will improve our understanding of already exciting results.

Line 106 and 118, Was there a specific reason to change the AnI incubation time from 3 hours to 5 hours?

Lines 166-171, Fig. 3 and Supl. Fig. 3 show that the membrane exposure or protein transfer for GSK-SrfB/E is much weaker than other samples. Can authors confirm this? In addition, have the authors noticed whether the *R. parkeri* mutants expressing GSK-tagged Srfs replicate at a comparable rate? Did the expression of SrfB/E cause any unforeseen activities in *R. parkeri* growth?

Lines 174-178, With the majority of SrfC detected in the pellet, is it possible that SrfC is released from the bacterial surface during replication or from dead bacterial bodies (rather than through an active secretion mechanism)?

Lines 182-185, Unlike SrfD and SrfF, immunofluorescent signals for SrfC were absent from neighboring cells seemingly infected with comparable levels of *R. parkeri*. The authors mentioned that the SrfC perinuclear staining was a rare event. How often have the authors observed the SrfC-positive signals in *R. parkeri*-infected cells?

Lines 266-269, it is difficult to assess the similarities and differences in the colocalizations of FLAG-tagged SrfD variants with Sec61 β . Does the white-colored area indicate those with overlapping signals?

Reviewer #3 (Remarks to the Author):

I have carefully reviewed the manuscript entitled "Cell-selective proteomics reveal novel effectors secreted by an obligate intracellular bacterial pathogen" submitted for my review. The authors have undertaken a commendable effort to address an important question in host-pathogen interaction, and the technical aspects of the work demonstrate a strong foundation. However, it is with regret that I recommend rejecting the manuscript in its current form.

I find it necessary to express reservations regarding the claimed novelty of the methodology employed. The authors appropriately reference previous works (references 25 to 28) demonstrating cell-selective BONCAT using azidonorleucine (AnI) in various bacterial pathogens, establishing this method as well-established. While the authors assert novelty in applying this

method to the obligate intracellular bacterial pathogen, *Rickettsia parkeri*, they justify their claim by highlighting challenges in cultivating *Rickettsia* spp. axenically. However, this argument is potentially misleading in the current context, as the authors themselves have successfully propagated different *Rickettsia* strains in host cell cultures through transformation with relevant plasmids, including the one expressing a mutant methionyl-tRNA synthase accommodating AnI. Therefore, the purported novelty in methodology is a mere extension of a well-established technique to a different bacterial pathogen (e.g. as reported by Franco, M. et al. *Front. Cell. Infect. Microbiol.* 8, (2018)). I recommend a thorough reassessment of the manuscript's significance in light of these considerations.

Fig. 1e: AnI-labeled proteins were detected via Western blot for biotin followed by tagging with alkyne-functionalized biotin. In the supernatant, four protein bands were labeled as putative secreted effector proteins. What is the basis of this assumption? A comparison with the corresponding lane from the pellet sample is not a good one because the two samples likely have different total protein loading, which could affect the electrophoretic mobility of proteins, and a single band may contain multiple proteins. As per lines 488-489, equal volume of pellet lysate was used as input for click reactions in the right side panel. This will certainly mean different input protein amounts in the left side and right side panels.

Are there no endogenous biotinylated proteins in *Rickettsia parkeri*? The first three clean lanes in the pellet samples indicate so.

Fig. 2: The entire premise of this experiment is based on the assumption of selective lysis of the host cells. I would like to see convincing data showing that the bacteria inside the host cells do not lyse under the "selective" lysis of the host cells. The fact that the so called "putative secreted rickettsial factors" (SrfA-G) identified are not located proximal to either type IV or type I secretion system components and are distributed across the species genome call for explanation. The concern here is what if there is a fundamental flaw in the selective lysis protocol? In this case, the detected proteins could simply be abundant proteins rather than secreted/effector proteins detected by the mass spectrometer. For validating their findings, the authors generated *R. parkeri* strains expressing the Srfs with glycogen synthase kinase (GSK) tags and infected Vero host cells. In principle, upon secretion into the host cytoplasm, the GSK-tagged proteins may be phosphorylated by host kinases, and in this case can be detected by immunoblotting with phospho-specific antibodies. But in addition to the fundamental problem associated with the overexpression of protein targets (therefore, it doesn't explain whether or not the targets studied are true physiologically relevant effectors) the validity of this experiment depends on the reliability of the antibodies used. Based on the Western blot images of Fig. 3a, although the chosen positive and negative controls gave expected results, I am not convinced whether the "phospho-specific" antibodies of the Srfs are truly specific for phosphorylation or not, and I am not convinced this experiment proves anything. Besides, the fact that SrfB and SrfE were not detected (and SrfG and SrfF detected with weak bands) make findings from this validation experiments inconclusive at this stage.

Fig. 5: Immunoprecipitation of endogenous SrfD from WT *R. parkeri*-infected host cytoplasmic lysates followed by mass spectrometry was performed to identify potential binding partners of the bacterial protein in the infected host cell. This experiment identified Sec61 α and β proteins from the host cell as potential binding partners of SrfD. However, the biological relevance of this interaction remains unknown and the localisation of SrfD to the endoplasmic reticulum does not seem to be affected by its interaction with the Sec61.

Overall, the findings presented, while intriguing, necessitate further validation through additional experiments to strengthen the impact of the study. It is my belief that these additional experiments are imperative to substantiate the conclusions drawn. I appreciate the authors' contributions to the field and encourage them to consider the suggested enhancements before resubmitting to ensure the manuscript reaches its full scientific potential.

Response to Reviewers' Comments

Reviewer 1:

General comments: The manuscript titled “Cell-selective proteomics reveal novel effectors secreted by an obligate 1 intracellular bacterial pathogen” by Sanderlin et al. uses a novel approach to Rickettsiology (BONCAT) for identifying effectors that are secreted by *Rickettsia parkeri* during host cell infection. BONCAT (bioorthogonal non-canonical amino acid tagging) utilizes the need for rickettsia to uptake amino acids from the host environment by introducing a tagged amino acids that are derivatives of expressed and translocated effectors. The approach captured validated and novel effectors, with further experimentation on several novel effectors showing they localize to the host cytoplasm, mitochondria, and ER. One novel secreted rickettsial factor, SrfD, was further determined to interact with the host Sec61 translocon.

Overall, this is an excellent report that will not only be hugely impactful to Rickettsiology, but also inspire pathogenesis work on fastidious microbes that are difficult to study. The approach is clever (yet practical) and will certainly be utilized by the field for related microbes. The identification of novel effectors with experimental evidence is a big contribution to the field. This is a study from a younger PI who is steadily making significant contributions to the field of Rickettsiology, with this specific achievement unearthing insight on how rickettsiae exploit host cells to facilitate invasion, replication and intercellular spread. Clearly there is some connection with *Rickettsia* infection and targeting the host secretory pathway at the ER. The PI also has a paper on BioRx demonstrating that *R. parkeri* interacts with the ER envelope, which complements the findings here for interactions between SrfD and the the host Sec61 translocon.

I find several shortcomings with this otherwise important and well-conducted study. First, the literature review of rickettsial protein secretion is very limited. The author cites two important works (an earlier secretome review by Gillespie and colleagues) and an unpublished work by the same group (Verhoeve et al.) that contain a lot of findings and literature that should be better captured in this report. Some specific details on this oversight are mentioned below. Second, the phylogenomics analysis is very limited relative to the dozens of diverse *Rickettsia* and *Tisiphia* genomes now available. This makes the paper very *R. parkeri*-centric, with minimal comparative power and overall applicability to the genus *Rickettsia* as a whole.

Response: We appreciate the reviewer's thorough review of the work and we understand their desire for more discussion of rickettsial secretion and comparative phylogenomics. Below, we address the specific requests in detail. Overall, we agree that the prior phylogenomic analyses are important foundations for the field, especially for generating hypotheses about candidate secreted effectors that should then be experimentally validated during *Rickettsia* infection. The major goal of this study was not to predict effectors but to isolate effectors delivered into the host cell, followed by multiple levels of validation. For these reasons, and considering the technical challenges in the *Rickettsia* field, we chose to perform all experiments in *Rickettsia parkeri*, which has proven to be

an excellent model system for accelerating technological innovations and recapitulating many aspects of spotted fever pathogenesis. We completely agree that this work should motivate comparative analyses across the genus, and we are eager to address these kinds of questions in subsequent work where a large-scale *in silico* analysis would be more appropriate.

Specific comment #1: Naming novel effectors Srfs may pose a problem down the road. A nomenclature already exists for rvh T4SS effectors (REMs) and some of the Srfs could be secreted by the rvh T4SS, so they would also qualify as REMs. Since this study characterized one species, *R. parkeri*, it may be more prudent to name the seven Srfs something more specific to this species, as has been done for *Anaplasma* T4SS effectors when annotations on protein function cannot be made. This is just a consideration. “Secreted” also really applies to surface localized proteins.

Response: No T4SS-null mutant exists in these bacteria to experimentally validate that these are indeed T4SS effectors. Thus, it would be inappropriate to use the REM nomenclature for these proteins. Further, we followed the current guidelines set forth by the American Society for Microbiology, which states that “gene names should not begin with prefixes indicating the genus and species”. We therefore named the Srfs based on their shared behavior of being secreted into the host cell milieu, which we demonstrate by selective lysis, reporter fusions, and/or microscopy-based approaches. While the surface proteome plays a crucial role in the rickettsial life cycle, as discussed below, we clarify that the goal of our study was to identify the subset of secreted proteins that are delivered into the host cell milieu.

Specific comment #2: The Abstract is nicely written. However, providing numbers for known and novel effectors would be helpful. For instance, how many effectors identified by BONCAT have some other experimental support for being secreted (by any *Rickettsia* species)? Of the novel effectors, have any been captured in screens from other papers (those using biochemical approaches to identify secreted proteins)? As written, it seems like there are many more effectors than there actually are, but without listing the exact contribution to the effector landscape, the readers will really have no idea what gains this work has accomplished.

Response: For the purposes of this work, we focused on the subset of secreted proteins that are experimentally demonstrated to be delivered into the host cell milieu during *Rickettsia* infection (using *e.g.*, microscopy, reporter fusions, or selective lysis approaches). In that light, we count Pat1, Pat2 (absent in *R. parkeri*), Risk1, RaIF (absent in *R. parkeri*), RARP-2, and Sca4 as known effectors secreted into the host cell. Thus, the experimental identification of the seven Srfs more than doubles the number of these types of proteins known for *R. parkeri* in particular and for the *Rickettsia* genus in general. RickA has been shown to associate with the rickettsial surface, and therefore is not part of the subset of secreted proteins we focused on in this study (see below for more on this topic). *R. felis* VapC was identified in the infected host cytoplasm only upon treatment with chloramphenicol by Audoly *et al.* (PMID 22046301), and the authors speculated that detection of VapC under these circumstances could represent protein

released during bacterial death (rather than active delivery into the host cell). Delivery of other candidate effectors (e.g., cREMs, Pld, TlyA/C, Risk2) into the host cell has not yet been demonstrated during *Rickettsia* infection. We have clarified our use of the term “secreted” in the text as follows:

Lines 42–45: “Despite these advances, however, the subset of secreted proteins that *Rickettsia* spp. deliver into the host cell milieu to drive infection has remained elusive; recent studies have characterized only a handful of such secreted rickettsial factors.”

Lines 51–55: “Aside from these six experimentally validated effectors, however, the effector arsenals of *Rickettsia* spp. remain a mystery. Given that other bacterial pathogens secrete dozens if not hundreds of effectors into the host cell^{20–24}, there is a pressing need to identify new rickettsial effectors.”

Prior to our work, none of the SrfA homologs had been shown to be secreted into the host cell by any *Rickettsia* species or found to interact with Rvhd4 in co-IP proteomic studies. As we noted in the text (lines 385–389), however, the *R. conorii* SrfA homolog RC0497 is found abundantly in infected cell culture media and infected host sera (Zhao *et al.* [PMID 31955791]) and had been observed in the periplasm of purified rickettsiae by immunogold labeling (Patel *et al.* [PMID 31299006]). Neither of these works tested for secretion of RC0497 into the host cell, although the detection of extracellular RC0497 by Zhao *et al.* is consistent with rickettsial secretion into the host cell and subsequent release by the host cell during infection.

Specific comment #3: Line 53-60: This really falls short of a synopsis for the protein secretion research on rickettsiae. The authors cite a recent paper (Verhoeve *et al.*) that lists many more bona fide and candidate T4SS effectors. Another paper from the same group that the authors also cite (Gillespie *et al.*) covers all secretory proteins in greater detail, including the work from Azad that analyzed the SEC signals of many Sec-translocated proteins. Still, there are several works that use tags or whole protein antibodies to show secretion during host cell infection. Some of these studies have used B2H and/or Co-IP/pull-down to show effector interactions with either host molecules or secretion machine components. All this literature cannot go undiscussed. As written, this section is a highly inaccurate depiction of the status of the field.

Response: The focus of this report was specifically on rickettsial factors that are secreted into the host milieu, excluding proteins that are displayed on the surface or access the periplasm (e.g., clients of the Sec machinery). We appreciate how our use of the term “secreted” could cause a misunderstanding and have clarified the scope of our study in the manuscript (see changes in lines 16–17 and 42–44).

Per the reviewer’s request, however, we have also amended the text to acknowledge the pioneering work of other groups to identify rickettsial surface proteins and Sec clients as follows:

Lines 37–42: “Extensive efforts to characterize proteins secreted to the rickettsial surface have revealed unique ways that these bacteria interact with host cell machinery. For example, the major outer membrane proteins OmpA and OmpB

mediate host cell invasion^{4,5}, and the surface proteins Sca2 and RickA polymerize actin to drive motility within the host cytoplasm⁶. Furthermore, biochemical studies have identified myriad surface proteins that could likewise support the rickettsial life cycle^{7–11}.”

We have also made changes to explicitly address prior work using bioinformatic approaches and interaction assays to identify candidate T4SS effectors as follows:

Lines 74–83: “Through two-hybrid and co-immunoprecipitation approaches^{13,17,18}, a series of *rvh* effector molecules (REMs) has been identified based on interactions with the *Rickettsiales vir* homolog (*rvh*) type IV secretion coupling protein RvhD4³⁷. Bioinformatic analyses have highlighted additional candidate REMs by virtue of their similarity to existing REMs, but secretion for many of these proteins has not yet been experimentally validated. Furthermore, interactions with RvhD4 are not conclusive proof of secretion because many of the other proteins that co-immunoprecipitate with RvhD4 include housekeeping proteins that are presumably not secreted into the host cell¹⁷. Unfortunately, the lack of a secretion-null *Rickettsia* mutant precludes validation of any effector as a true *rvh* substrate.”

We note that while the results of these latter approaches can generate important hypotheses, they do not demonstrate effector delivery to the host cell during rickettsial infection. Furthermore, our experimental approach to identify effectors is secretion system-agnostic.

Specific comment #4: Line 75-82: Again, not stating clearly what the overall contribution of the work is (numbers of newly identified effectors) makes the significance hard to realize.

Response: In addition to the changes noted above, as well as lines 164–165 present in the original manuscript, we more clearly state the number of new effectors (seven) in the abstract as follows:

Lines 19–20: “The seven novel secreted rickettsial factors (Srfs) we identified include *Rickettsia*-specific proteins of unknown function that localize to the host cytoplasm, mitochondria, and ER.”

Specific comment #5: Demonstrating MetRS* expression does not impede cell-to-cell spread or bacterial growth, or seemingly overall expression or bacterial fitness, is appreciated.

Response: We thank the reviewer for recognizing this important control.

Specific comment #6: The MASS SPEC results seem very clean. Is it standard to not include all peptides from the analysis? Check with the journal requirements. Also, some of the proteins listed as having minimal hits are known to be secreted (RickA and Sca5). Sca5 in particular is probably present more than any other protein on the surface of rickettsial cells and is also processed. It is also highly expressed, so AnI incorporation should be very high. RickA is well characterized by the Welch group and others and is

active during *R. parkeri* infection. Why are these proteins not labeled with AnI like the other effectors?

Response: As described in the Methods and Data Availability sections, Data Set 1 (BONCAT pull-down) peptide identifications were made against the *R. parkeri* str. Portsmouth proteome and Data Set 2 (SrfD co-IP) peptide identifications were made against the *R. parkeri* and human proteomes. We used a standard cutoff of two unique peptides for identification of proteins in both approaches, which we have clarified in the text as follows:

Lines 768–770: “Protein identifications were accepted with two unique peptides, and normalized intensities from the top three precursors were computed with Scaffold (Proteome Software).”

Protein IDs, intensities, spectral and peptide counts, and percent coverage from these searches are all included in the Data Sets. The full mass spectral data are available on the public proteomics repository MassIVE (<https://massive.ucsd.edu>, MSV000093380 and MSV000093381) in compliance with Springer Nature guidelines.

As mentioned in the Discussion section (starting at line 455), the timing of AnI labeling, efficiency and tolerance of AnI incorporation, timing and extent of effector production and secretion, efficiency of effector extraction and processing, and peptide detection all impact hit identification from the BONCAT pull-down approach. These caveats aside, we note that Sca5/OmpB and RickA are surface-bound and therefore not likely to be BONCAT hits unless they are cleaved off into the host milieu during infection (as we detected peptides from the passenger domains of Sca1 and OmpA). As the reviewer states, Sca5/OmpB is one of the most abundantly-expressed proteins and yet very few peptides were detected (and all mapped to the Sca5/OmpB passenger domain); this result provides further evidence that our selective lysis protocol causes minimal disruption to the bacteria and enables extraction of secreted effectors delivered into the host cell. The work by Reed *et al.* (PMID 24361066) showed that, while RickA is active during infection as noted by the reviewer, it is largely absent from the *R. parkeri* surface at 48 h post-infection; our late labeling window would therefore further limit detection of RickA in combination with the aforementioned limitations.

Specific comment #7: FIGURE 1: Nicely done and convincing that BONCAT was successful. For panel C, the bacteria should be further described as purified or partially purified, no?

Response: We thank the reviewer for their assessment of the BONCAT proof-of-concept experiments. Fig. 1c is counting the number of bacteria in each infectious focus while inside infected cells (rather than purified, isolated rickettsiae). Please see lines 549–551 in the Methods section for experimental details.

Specific comment #8: FIGURE 2: the cladogram depicting the phylogeny of the major Rickettsia groups does not agree with current robust genome-based phylogeny estimations. The TRG and TG are monophyletic by the works of Gillespie, Ettema, and

Hurst and their teams, and their phylogeny estimations are very robust. This is also a poor representation of the actual *Rickettsia* diversity, but it may not be the goal of the authors to be thoroughly robust yet just list a couple examples. AG, is an old term and does not include any natural monophyletic group. BG refers to the bellii group, which now has nearly a dozen genomes. But *R. canadensis* is not a member of AG. The authors could just remove the cladogram and AG and the graphic would be fine. Or redraw the tree and leave out AG.

Response: We thank the reviewer for their suggestions and have removed the cladogram and group names from this figure. We have also included additional *Rickettsia* spp. in the figure to showcase more of the diversity of Srf conservation across the genus. Future work cataloging Srf homologs across every species in the genus will be important, but we hope the reviewer agrees that these analyses are not required to support the conclusions of our work and are beyond the scope of this study.

Specific comment #9: Of the seven Srf proteins, how many have other experimental evidence for expression, secretion, subcellular localization, mutagenesis, etc. from studies in the literature? What about their homologs in other *Rickettsia* species? How about the best BlastP results against non-*Rickettsia* organisms? In current form, there is not much information provided from the bioinformatics analyses that were conducted.

Response: As discussed above, this work is the first to provide experimental evidence for secretion of the Srfs into the host cell. This work is also the first to demonstrate the subcellular localization of the exogenously-expressed Srfs and, with the exception of SrfA, the first to investigate the localization of the endogenous proteins. As discussed above, prior work demonstrated that the *R. conorii* SrfA homolog RC0497 is abundantly expressed during infection. Some of the remaining Srfs have appeared as hypothetical proteins in prior published datasets, but without follow-up work. Thus, the Srfs remain largely uncharacterized. We have included references to these studies in the text as follows:

Lines 428–436: “Additionally, a thorough characterization of Srf expression could clarify their roles across the genus and across host cell niches. For example, *R. conorii* *srfB* is highly transcribed during infection of human and tick cells⁸², and the expression of *R. rickettsii* *srfE* and *R. typhi* *srfG* appears to respond to temperature shifts^{83,84}. We eagerly await the generation of *srf* mutants, the study of which will provide insight into host these effectors contribute to the rickettsial lifestyle and pathogenesis. Transposon mutants for *srfB* from *R. parkeri* and *srfF* from *R. prowazekii* and *R. felis* have been isolated but remain uncharacterized^{85–87}. Altogether, the results from these future studies will offer a more comprehensive view of Srf biology.”

As mentioned in the manuscript (lines 165–167 and 385–387), SrfA is a predicted amidase and has homology to non-*Rickettsia* amidases; for the remaining Srfs, all but SrfD lack homologs outside the *Rickettsia* genus and thus appear to be *Rickettsia*-specific. A BLAST search with full-length SrfD does yield non-*Rickettsia* hits in diverse organisms, but these align to the low-complexity pentapeptide repeats (PPRs) of SrfD;

notably, repeating this search using the regions of SrfD outside the PPRs results in zero non-*Rickettsia* hits. We have clarified the text addressing the reviewer's question about conservation outside the genus as follows:

Lines 167–170: “The hypothetical protein SrfD has partial sequence homology to uncharacterized pentapeptide repeat-containing proteins in diverse taxa, but only *Rickettsia* spp. encode homologs of full-length SrfD. The remaining Srfs are hypothetical proteins with no sequence homology outside the *Rickettsia* genus.”

Lines 823–824: “Protein-protein BLAST was also used to detect putative Srf homologs in organisms excluding those of the *Rickettsia* genus.”

Specific comment #10: Also, entering the locus tags into NCBI does not return any data. The authors should use the current GenBank accession numbers so that the reader can access the sequences without problems. They state that the Portsmouth strain was used but the accession numbers for the Srfs are not provided. The WP_ numbers in Supplemental Table 1 can be used to trace down the proper locus tags or individual accession numbers. For instance, MC1_RS02555 is the locus tag MC1_02805 and accession number AFC74691.1. These numbers would be better in Figure 2 than those listed. Unless I am missing something.

Response: We thank the reviewer for their suggestion. We note that, following the RefSeq prokaryotic genome re-annotation project, the MC1_RS# locus tags for *R. parkeri* str. Portsmouth are the current locus tags (and the MC1_# locus tags are outdated). To facilitate future searches, however, we have replaced the locus tags in Figure 2 with the *R. parkeri* str. Portsmouth protein accession numbers as suggested; these accession numbers have also been added alongside the non-redundant WP_# accession numbers in Data Set 1.

Specific comment #11: Line 145-146: SrfG has been called cREM-2a in the cited preprint by Verhoeve et al.

Response: We have included a reference to this nomenclature in the results section as follows:

Lines 181–183: “In a recent bioinformatic analysis³⁷, SrfG was nominated as a candidate REM (cREM-2b) but had not been validated as a secreted effector or RvhD4 interaction partner in that work.”

We note that SrfG was referred to as cREM-2b in the aforementioned paper, with cREM-2a being the downstream DUF5410-containing paralog found as an interaction partner of RvhD4.

Specific comments #12 and #13: Line 148: Is there any information from genome synteny or gene neighborhood across *Rickettsia* species that can link some of the Srfs to other proteins that may be involved in host interactions? Are these gene neighborhoods conserved across species or variable (e.g., is there any evidence for recombination hotspots, pseudogene graveyards, etc.)? Supplemental Figure 2 is not very helpful for

understanding gene neighborhood or relative conservation of Srf neighborhoods across *Rickettsia* genomes. The BVR-BRC compare region viewer tool could help here.

SrfA-D are absent in *TG rickettsiae* and overall seem to show a patchwork distribution across the remaining *Rickettsia* genomes. It might be nice to expand on this regarding pseudogenization in *Rickettsia* genomes and how lineage specific factors “come-and-go” via LGT and gene decay.

Response: We thank the reviewer for their helpful suggestion of the BV-BRC tool. The genome diagram in Supplementary Fig. 2 was included to emphasize that the *srf* loci are scattered across the genome and are not located proximal to genes encoding components of the T1SS or T4SS. Per the reviewer’s suggestion, we have included an additional panel (Supplementary Fig. 2b) that displays the gene neighborhoods for each *srf* across several *Rickettsia* genomes. We believe that a thorough exploration of *srf* evolution and diversification would be a valuable direction for future work in the field, but it is beyond the scope of the current study and not necessary to support the conclusions of this work. We have included some of the observations from this suggested analysis in the revised manuscript as follows:

Lines 207–217: “The *srf* gene neighborhoods are largely conserved across the *Rickettsia* genus (Supplementary Fig. 2b), and the flanking genes are often intact even in species where a particular *srf* is fragmented or absent. Furthermore, with the exception of the *srfG* and *cREM-2a* gene pair encoding DUF5410-containing proteins³⁷, there is no obvious functional link between the *srf* genes and the conserved flanking genes. In contrast to their secreted effector neighbors, the proteins encoded by these flanking genes include those involved in housekeeping functions like DNA repair and recombination (e.g., XerD, RadA, and RecO), tRNA and rRNA modification (e.g., TsaB, RluB, RsmD, and MnmE), translational initiation (InfB), and peptidoglycan processing (Slt and IdcA). Altogether, these findings motivate a more comprehensive analysis of *srf* evolution and diversification in future work.”

Lines 421–428: “Although the Srfs are unevenly distributed across the genus, the genomic regions surrounding the *srf* loci are well-conserved; this suggests that the *srf* genes and their flanking genes may face different selective pressures. The presence and diversification of these unique effectors in bacteria with notoriously streamlined genomes raises exciting questions about rickettsial evolution within the host cell niche⁷⁸. Bioinformatic studies have traced the emergence, maintenance, and decay of genes across *Rickettsia* spp.^{37,79–81}, and the Srfs could serve as useful focal points for such analyses in future work.”

Specific comment #14: For SrfD, the NCBI CDD returns a match to “type IVB secretion system protein DotG/IcmE”. This is a scaffold protein that utilizes the pentapeptide motif to interact with other scaffold components within the I-T4SSs. This may indicate a function for SrfD in building a scaffold for translocation across the ER membrane. This could be worth mentioning in the discussions on SrfD-Sec61 interactions.

Response: We thank the reviewer for their suggestion. We have elaborated on the SrfD PPRs in the text as follows:

Lines 369–372: “PPRs from diverse proteins make direct contacts with binding partners or otherwise serve as rigid scaffolds for other interaction domains⁵⁵. Whether the SrfD PPRs directly interact with Sec61 or serve as spacers to position CC1 for such an interaction remains to be determined.”

Specific comment #15: SrfA is a predicted PGN hydrolase and the *R. conorii* homolog that has that activity; why, then, is SrfA secreted? Does it not act on *R. parkeri* PGN but PGN of congener microbes (interbacterial antagonism)? This protein likely doesn't reach the periplasm (although the cited paper says otherwise), yet probably functions equivalent to AmpD, which degrades PGN subunits in the cytoplasm after they are imported via AmpG during murein turnover. Regardless, what would its target be in the eukaryotic cytosol? Moonlighting?

Response: We agree with the reviewer that these are all feasible models. We felt that it was overly speculative to discuss potential roles of SrfA in the current study, especially given word limit constraints. We hope to explore these ideas in future studies, however, as we can also speculate a model wherein SrfA could help limit activation of cytoplasmic peptidoglycan sensors in the event that peptidoglycan is released during infection.

Specific comment #16: Line 154: Indeed! Major point of the paper here.

Response: We thank the reviewer for their positive comment.

Specific comment #17: Line 169: Is there any information for SrfB and SrfE expression/secretion from the literature?

Response: As discussed above, this work is the first to provide experimental evidence for Srf secretion, and existing information about Srf expression from the literature is limited.

Specific comment #18: Line 174: Why were antibodies not raised against SrfA and SrfG? Since they were expressed, why not determine their subcellular localization?

Response: We prioritized production of antibodies against effectors for which we could successfully generate recombinant proteins. During review of this manuscript, we generated antibodies against SrfE and demonstrated its secretion (Fig. 3c,d), which we have now included in the manuscript. Further work is needed to generate specific antibodies against the remaining effectors.

Specific comment #19: Line 187: It might help to distinguish the host cell cytoplasm from the bacterial cytoplasm by calling the former “cytosol” and the latter “cytoplasm”. It can be confusing otherwise, or just use the proper adjective when using “cytoplasm”.

Response: Because cytosol refers to the fluid components of the cytoplasm, it would be confusing to the readers if we attempt to redefine such cross-disciplinary terms. We have therefore altered the language in the manuscript as follows:

Lines 262–264: “Finally, we noted staining for SrfF in the infected host cytoplasm, the intensity of which similarly increased at higher bacterial burdens.”

Specific comment #20: Line 201: Along these lines, should it be stated that ectopic expression sans infection will dissociate the secretion pathways for Srf, which may be important for the accurate delivery to their intended host targets. So, the observed subcellular localization patterns might be different if the same experiments were done during infection.

Response: We thank the reviewer for their helpful suggestion. We have expanded our comparison of the endogenous and exogenous Srf localization patterns as follows:

Lines 402–405: “These results suggest that infection-specific cues dictate effector localization or, more simply, that exogenous expression disassociates secretion of a Srf from its delivery to a particular subcellular compartment.”

Specific comment #21: Line 212: The experimental evidence for the DrfD-Sec61 interaction is robust.

Response: We thank the reviewer for their critical assessment of our interaction data.

Specific comment #22: The Discussion is nicely written.

Response: We thank the reviewer for their positive comment on our discussion.

Specific comment #23: Line 329: The most current reference for this system is PMID: 27307105.

Response: We thank the reviewer for their suggestion and have added this reference.

Specific comment #24: Line 332-334: The signals for most T4SS effectors have been shown to be genus- or species-specific, and they are typically confirmed by experimental assays. Just because programs exist to predict non-SEC secreted effectors does not mean any of them are decent or trained on anything conserved or universal (unlike the SEC and TAT signals that are highly conserved and amenable to bioinformatics tools). The way this is written implies that other organisms have in silico tools that work for predicting non-SEC secreted proteins. This is simply not true. And the algorithms do not perform well on other species when they do show some promise for one particular species (like a workhorse that has a lot of effectors experimentally characterized, so the models and algorithms work in that particular system).

Response: We completely agree with the reviewer about the limitations of T4SS prediction algorithms. We have clarified this point based on the reviewer’s suggestion:

Lines 441–444: “Even for well-studied pathogens, however, the signal sequences for substrates of these secretion systems are not universal, and often they are genus- or species-specific^{36,89–91}; thus, robust computational prediction of effectors secreted by *Rickettsia* spp. is challenging.”

Specific comment #25: Line 336-342: the named SrfG effector (AFC74686) contains the DUF5410 domain that is unique to *Rickettsia* species. Verhoeve et al. (bioRxiv. 2023 Feb 27:2023.02.26.530123. doi: 10.1101/2023.02.26.530123. Preprint) recently analyzed these proteins and found them to belong to a multi-gene family (Figure 7b in that report). The authors cite this work, but the effectors were named cREM-2 with letters denoting their distinct groups based on phylogeny estimation and domain analyses. It would be helpful for the community and literature if the “cREM2” nomenclature were used or at least referenced. This could also be an opportunity for the authors to take a stab at mentioning the “REM” and “cREM” annotations from Verhoeve et al. and how their “Srf” annotations fit in. It would be great to acknowledge the work from both groups and call for consensus. Also, the fact that cREM-2a of *Rickettsia typhi* (locus tag RT0352) bound RvhD4 in a pull-down assay with *R. typhi* lysate, yet the adjacent gene encoding the cREM-2b protein (locus tag RT0351) accentuates that all of these experimental approaches are not “all-or-none”. The pull-down from the Azad lab papers that captured REMs and cREMs interacting with RvhD4 missed one of the effectors captured by Sanderlin et al., RARP-2, which has not only been shown to bind RvhD4 in one-to-one Co-IPs and B2H assays, but that interaction is abolished when the C-terminal RARP-2 tail is removed (Lehman et al., PMID: 29946049). It would be good for the authors to acknowledge that all of these experimental methods have limitations, and that results across different labs using different species and strains may not corroborate one another, but any experimental data nonetheless is important to call out.

Response: As discussed above, we have now included references to the REM nomenclature. We maintain that the Srf nomenclature is most appropriate given that these proteins were identified for being secreted into the host cell and that no experimental evidence yet exists for their secretion by the T4SS. We agree with the reviewer’s suggestion and have included a more explicit mention of the prior T4SS interaction data (discussed above). Interactions with RvhD4 may provide candidates for secretion by the T4SS, but they are not definitive proof. Relatedly, failure to detect an interaction with RvhD4 from a particular species, as the reviewer correctly noted, does not rule out that an effector is a T4SS substrate and secreted into the host cell. Indeed, the *R. typhi* RvhD4 crosslink-based co-IP by Voss *et al.* did not identify any of the Srf homologs encoded by *R. typhi* (SrfE/F/G) as interaction partners.

Specific comment #26: Greater depth with phylogenomics could help discern those Srfs present in bona fide non-pathogens like *R. peacockii*, REIP, and *R. buchneri*. Based on the current presentation, it isn’t clear if any of the Srfs can possibly define interactions with vertebrate cells and not both vertebrate and arthropod cells (the non-pathogens wouldn’t need Srfs if they are only required for vertebrate cell takeover).

Response: We thank the reviewer for their suggestion. We have included additional *Rickettsia* spp. in Fig. 2 to showcase more of the diversity of Srf conservation across the genus, including among known pathogens and non-pathogens as mentioned by the reviewer. We have expanded our discussion of Srf conservation in the text as follows:

Lines 185–192: “The Srfs are variably conserved within the *Rickettsia* genus. For example, homologs of SrfE are found across the genus, whereas SrfB homologs are only present in a subset of species. Some Srf homologs are fragmented (e.g., *R. felis* SrfB) or otherwise highly divergent from the *R. parkeri* Srf (e.g., *R. typhi* SrfF), suggesting that Srf function is not shared between all species. Furthermore, the presence or absence of a given Srf homolog appears to be independent of pathogenicity: pathogenic (c.f., *R. rickettsii* str. Sheila Smith and *R. typhi*) and non-pathogenic (c.f., *R. peacockii* and *R. buchneri*) species alike encode either full sets of Srf homologs or are missing particular Srfs.”

While a deeper phylogenomic analysis would generate many excellent hypotheses, it is not required to support the conclusions of our work identifying effectors secreted into the host cell milieu during infection. We hope the reviewer agrees that such an analysis would be more appropriate in the future and we are eager to collaborate with the experts in the field to examine these questions.

Specific comment #27: Methods seem fine. Remaining figures and supplements are fine.

Response: We thank the reviewer for their comprehensive assessment of the manuscript.

Reviewer 2:

General comments: In this manuscript, Sanderlin et al. successfully demonstrated the application of BONCAT for *R. parkeri*, a tick-transmitted obligate intracellular pathogen, and identified novel molecules deposited in various subcellular compartments during *R. parkeri* replication in mammalian host cells. This is a huge step forward for the field where traditional genetic and bioinformatic tools demonstrate low efficiencies in identifying novel factors and characterizing their molecular functions impacting host-pathogen interactions. The manuscript is well-written, with supportive data and detailed descriptions of experimental methods. Below are a few questions and comments that will improve our understanding of already exciting results.

Response: We thank the reviewer for their critical assessment of our work and its impact on the field. Below, we address the specific comments in detail.

Specific comment #1: Line 106 and 118, Was there a specific reason to change the AnI incubation time from 3 hours to 5 hours?

Response: Although we found that 3 h AnI incubation was sufficient to detect labeled proteins by immunoblotting, we increased the incubation period for the BONCAT pull-down to maximize the time for effector labeling and secretion (and ultimately increase the chance of detection by mass spectrometry).

Specific comment #2: Lines 166-171, Fig. 3 and Supl. Fig. 3 show that the membrane exposure or protein transfer for GSK-SrfB/E is much weaker than other samples. Can authors confirm this? In addition, have the authors noticed whether the *R. parkeri* mutants expressing GSK-tagged Srf proteins replicate at a comparable rate? Did the expression of SrfB/E cause any unforeseen activities in *R. parkeri* growth?

Response: We can confirm that GSK-tagged SrfB and SrfE were not detectably expressed: in a pilot blot for GSK-SrfB/E (included below), we note that these proteins were also not detectably expressed:

Western blots for GSK-tagged constructs expressed by *R. parkeri* during infection of Vero cells as in Fig. 3a. Whole-cell infected lysates were probed with antibodies against the GSK tag (bottom) or its phosphorylated form (P~GSK, top) to detect exposure to the host cytoplasm. Uninfected host cells and strains expressing GSK-tagged BFP (non-secreted) and RARP-2 (secreted) were used as controls. SrfB and SrfE (expected 37 and 50 kDa, respectively) were not detected. Right, western blots with enhanced contrast.

Nevertheless, we observed no obvious growth defects while generating or expanding these strains, and they were GFP-positive and spectinomycin-resistant, indicating that they successfully maintained the expression plasmid. We have included this observation in the manuscript as follows:

Lines 236–238: “The strains expressing GSK-tagged SrfB and SrfE were GFP-positive and spectinomycin-resistant, indicating that they successfully maintained the expression plasmid.”

Because all of the GSK-tagged Srfs exhibited variable expression despite being driven from the same promoter, we speculate that tagging SrfB/E altered expression or stability of these proteins. To circumvent this issue of expressing tagged SrfE, we successfully generated antibodies against endogenous SrfE during review of this manuscript. In the new Fig. 3c,d, we show that SrfE is secreted into the host cell. These results provide further validation that our cell-selective BONCAT approach can identify *bona fide* secreted effectors and underscore the importance of using multiple assays to confirm effector secretion. We have included these new data in the manuscript.

Specific comment #3: Lines 174-178, With the majority of SrfC detected in the pellet, is it possible that SrfC is released from the bacterial surface during replication or from dead bacterial bodies (rather than through an active secretion mechanism)?

Response: We appreciate the reviewer’s concern about SrfC secretion. The absence of RpoA in the supernatant fractions of Fig. 3b,c and the absence of phosphorylated GSK-tagged BFP in Fig. 3a suggest that adventitious release during division or lysis is not driving the release of the Srfs into the host cell. Furthermore, if SrfC were deposited onto the bacterial surface, we would expect to see staining of SrfC along the bacterial surface in Fig. 3d; instead, SrfC staining was limited to the host perinuclear region. Based on the different ratios of Srfs present in the supernatant vs pellet, we speculate that the Srfs may be secreted into the host cell with different efficiencies.

Specific comment #4: Lines 182-185, Unlike SrfD and SrfF, immunofluorescent signals for SrfC were absent from neighboring cells seemingly infected with comparable levels of *R. parkeri*. The authors mentioned that the SrfC perinuclear staining was a rare event. How often have the authors observed the SrfC-positive signals in *R. parkeri*-infected cells?

Response: We thank the reviewer for their helpful suggestion. We have clarified the frequency of these events in the text as follows:

Lines 257–260: “We observed rare instances (less than 3% of infected cells) of perinuclear staining for SrfC during infection, which was typically undetectable

even at higher bacterial burdens. SrfE behaved similarly with rare instances (approximately 5% of infected cells) of perinuclear staining.”

Specific comment #5: Lines 266-269, it is difficult to assess the similarities and differences in the colocalizations of FLAG-tagged SrfD variants with Sec61 β . Does the white-colored area indicate those with overlapping signals?

Response: Yes, the white areas indicate overlapping cyan (FLAG-SrfD) and red (Sec61 β) signals. We did not observe obvious differences in colocalization or ER morphology for these variants, and simply noted that all were found to localize to the ER. We have clarified our mention of the signal colors in the legend for Fig. 5d (as well as in the legends for Fig. 4b,c) as follows:

Line 1218: “White indicates overlap between FLAG and AIF signals.”

Lines 1221–1222: “White indicates overlap between FLAG and mNeonGreen signals.”

Lines 1237–1238: “White indicates overlap between FLAG and Sec61 β signals.”

Reviewer 3:

General comments: I have carefully reviewed the manuscript entitled “Cell-selective proteomics reveal novel effectors secreted by an obligate intracellular bacterial pathogen” submitted for my review. The authors have undertaken a commendable effort to address an important question in host-pathogen interaction, and the technical aspects of the work demonstrate a strong foundation. However, it is with regret that I recommend rejecting the manuscript in its current form.

Response: We thank the reviewer for their time critically assessing our work. Below, we address the concerns raised by the reviewer in detail.

Specific comment #1: I find it necessary to express reservations regarding the claimed novelty of the methodology employed. The authors appropriately reference previous works (references 25 to 28) demonstrating cell-selective BONCAT using azidonorleucine (Anl) in various bacterial pathogens, establishing this method as well-established. While the authors assert novelty in applying this method to the obligate intracellular bacterial pathogen, *Rickettsia parkeri*, they justify their claim by highlighting challenges in cultivating *Rickettsia* spp. axenically. However, this argument is potentially misleading in the current context, as the authors themselves have successfully propagated different *Rickettsia* strains in host cell cultures through transformation with relevant plasmids, including the one expressing a mutant methionyl-tRNA synthase accommodating Anl. Therefore, the purported novelty in methodology is a mere extension of a well-established technique to a different bacterial pathogen (e.g. as reported by Franco, M. et al. *Front. Cell. Infect. Microbiol.* 8, (2018)). I recommend a thorough reassessment of the manuscript's significance in light of these considerations.

Response: We agree with the reviewer that we extended the cell-selective BONCAT approach, and we appreciate the reviewer's comment that we diligently cited the prior foundational work using this technique in other organisms.

We would like to clarify that the work's major impact is the experimental discovery of seven novel secreted effectors that have eluded conventional methods of effector identification in the field. Without robust genetic tools to identify new secreted effectors, the field has often relied on bioinformatic approaches to predict candidate effectors. Such *in silico* strategies do not demonstrate effector secretion during infection and, as noted by Reviewers 1 and 2, they often perform poorly when adapted to more enigmatic pathogens like *Rickettsia*. Indeed, none of the SrfS were predicted as likely effectors by these tools. To address the reviewer's concerns, we direct them to the discussion items starting in lines 42, 66, 196, and 441.

Although multiple strains were generated by transformation in this work, the challenges of working with and manipulating these bacteria are a constant reality for the field. The lack of axenic culture for *Rickettsia* spp. (and other such obligate intracellular bacteria) is a well-known hurdle to studying rickettsial biology (as reviewed by McClure *et al.* [PMID 28626230] and Sit *et al.* [PMID 38315013]). Moreover, low transformation efficiencies

pose a considerable obstacle to generating rickettsial strains at scale (as reviewed by McGinn *et al.* [PMID 33784388]). These limitations demand the development of alternative tools to study *Rickettsia* spp. in greater detail, as our current understanding of these bacteria lags behind that of more tractable organisms. Altogether, our work bolsters our understanding of rickettsial biology and expands the sparse toolkit available for studying these bacteria. The impact of this development for the field was noted by both Reviewers 1 and 2.

Specific comment #2: Fig. 1e: AnI-labeled proteins were detected via Western blot for biotin followed by tagging with alkyne-functionalized biotin. In the supernatant, four protein bands were labeled as putative secreted effector proteins. What is the basis of this assumption? A comparison with the corresponding lane from the pellet sample is not a good one because the two samples likely have different total protein loading, which could affect the electrophoretic mobility of proteins, and a single band may contain multiple proteins. As per lines 488-489, equal volume of pellet lysate was used as input for click reactions in the right side panel. This will certainly mean different input protein amounts in the left side and right side panels.

Are there no endogenous biotinylated proteins in *Rickettsia parkeri*? The first three clean lanes in the pellet samples indicate so.

Response: We agree with the reviewer that the bands found in the supernatant versus pellet fractions are not comparable because they represent different locations in the infected system: the supernatant fraction would contain the infected host cytoplasm (and secreted effectors), whereas the pellet would contain intact bacteria. Since these fractions come from different compartments, we do not assume that total protein loading is equivalent between supernatant and pellet for a given condition. We have clarified in the text which lanes the reader should compare to observe the unique bands for the AnI-labeled, MetRS*-infected supernatant sample as follows:

Lines 132–137: “Consistent with our microscopy results, only the MetRS* strain exhibited appreciable labeling following treatment with AnI (Fig. 1e). Within this AnI-labeled, MetRS*-infected sample, the pellet fraction yielded a smear of bands, as expected for proteome-wide incorporation of AnI. Furthermore, the supernatant fraction contained several unique bands not found after infection with WT bacteria similarly treated with AnI (Fig. 1e, lane 4 versus lane 2).”

Yes, it is possible that endogenous biotinylated proteins exist for *Rickettsia parkeri* but, as the reviewer pointed out, they appear to be an insignificant population if present.

Specific comment #3: Fig. 2: The entire premise of this experiment is based on the assumption of selective lysis of the host cells. I would like to see convincing data showing that the bacteria inside the host cells do not lyse under the “selective” lysis of the host cells. The fact that the so called “putative secreted rickettsial factors” (SrfA–G) identified are not located proximal to either type IV or type I secretion system components and are distributed across the species genome call for explanation. The concern here is what if there is a fundamental flaw in the selective lysis protocol? In this case, the detected proteins could simply be abundant proteins rather than secreted/effector proteins

detected by the mass spectrometer. For validating their findings, the authors generated *R. parkeri* strains expressing the Srf proteins with glycogen synthase kinase (GSK) tags and infected Vero host cells. In principle, upon secretion into the host cytoplasm, the GSK-tagged proteins may be phosphorylated by host kinases, and in this case can be detected by immunoblotting with phospho-specific antibodies. But in addition to the fundamental problem associated with the overexpression of protein targets (therefore, it doesn't explain whether or not the targets studied are true physiologically relevant effectors) the validity of this experiment depends on the reliability of the antibodies used. Based on the Western blot images of Fig. 3a, although the chosen positive and negative controls gave expected results, I am not convinced whether the "phospho-specific" antibodies of the Srf proteins are truly specific for phosphorylation or not, and I am not convinced this experiment proves anything. Besides, the fact that SrfB and SrfE were not detected (and SrfG and SrfF detected with weak bands) make findings from this validation experiment inconclusive at this stage.

Response: We appreciate the reviewer's concerns about the selectivity of our lysis approach. Indeed, it is for this reason that we performed multiple orthogonal assays to confirm Srf secretion to the host cell in the absence of bacterial lysis. We have emphasized our use of multiple assays to demonstrate secretion in the text as follows:

Lines 265–267: "Altogether, the results from multiple assays – selective lysis, reporter fusions, and microscopy-based approaches – confirm Srf secretion into the host cell."

Lines 359–360: "Furthermore, we rigorously validated Srf secretion into the host cell milieu through multiple orthogonal assays."

As shown in Fig. 3b,c, the absence of a non-secreted control (RpoA) in the infected host cytoplasm fraction demonstrates that there is negligible lysis of bacteria that would confound detection of *bona fide* secreted proteins. Furthermore, the GSK tag assay (Fig. 3a) and immunofluorescence microscopy assay (Fig. 3d) do not rely on selective lysis and provide yet more evidence that the Srf proteins are secreted into the host cell. We refer the reviewer to prior studies by Nock *et al.* (PMID 35285700) and Sanderlin *et al.* (PMID 35727033) that demonstrate the outcomes of such assays when a protein is not in fact secreted into the host cell. The lack of phosphorylation of a non-secreted GSK-tagged control (BFP) demonstrates that there is negligible release of non-secreted proteins into the host cytoplasm for erroneous phosphorylation. Similarly, our ability to visualize Srf proteins in infected cells without permeabilization of the rickettsiae (e.g., with lysozyme and detergent) further demonstrates Srf secretion. We emphasize these points in the text as follows:

Lines 227–230: "Importantly, the lack of phosphorylation for GSK-tagged BFP demonstrates that there is negligible release of non-secreted proteins into the host cytoplasm during infection for erroneous phosphorylation."

Lines 243–246: "As shown previously⁴⁴, the bacterial RNA polymerase subunit RpoA was only detected in the pellet fraction, confirming that our selective lysis approach did not lead to adventitious rickettsial lysis that would confound validation (Fig. 3b)."

Lines 264–265: “Our ability to detect each of these proteins in the host cell without bacterial permeabilization further demonstrates Srf secretion.”

If adventitious bacterial lysis were responsible for our detection of the Srfs, we would anticipate that known high abundance proteins would be the top hits of our BONCAT pull-down approach. The surface protein OmpB is the most abundant protein expressed by *Rickettsia* spp., and yet OmpB was not robustly detected. The few peptides we did detect for this autotransporter protein map exclusively to the passenger domain which is known to be cleaved from the rickettsial surface (see work by Hackstadt *et al.* [PMID 1729180]), which may explain its detection alongside the passenger domains of Sca1 and OmpA. Moreover, high abundance *R. parkeri* cytoplasmic/non-secreted proteins (see work by Pornwiroon *et al.* [PMID 19797064]) were not robustly detected with our approach. Our inability to robustly detect such proteins compared to the Srfs reinforces that our lysis protocol is selective, an observation we have emphasized in the text as follows:

Lines 159–162: “High abundance internal rickettsial proteins were not robustly detected with this approach⁸, confirming that contamination of the supernatant fraction from adventitious bacterial lysis was minimal.”

As the reviewer correctly notes, the *srf* loci are not proximal to genes encoding components of the rickettsial secretion systems. Instead, they – like the previously identified effectors RARP-2, Risk1, and Pat1 – are scattered across the genome (see Supplementary Fig. 2a); even the genes encoding the T1SS and T4SS are themselves scattered across the genome, an observation made by others in the field (see work by Gillespie *et al.* [PMID 27307105]). Thus, it would be incorrect to assume that *Rickettsia* spp. effectors must lie close to components of the T1SS or T4SS. The discontinuous nature of these loci highlights the value of our proteomics-based approach to find novel secreted effectors that are not obvious from *in silico* studies of genome architecture. We emphasize this point in the text as follows:

Lines 196–198: “The fact that the *srf* loci are not obvious from studies of rickettsial genome architecture reinforces the value of experimentally identifying effectors secreted by these bacteria.”

We appreciate the reviewer’s concerns about the GSK tag secretion assay. As discussed above, this was only one of several orthogonal assays we used to demonstrate Srf secretion. As the reviewer correctly notes, this particular assay requires overexpression of tagged proteins. Even so, the GSK tag assay gave the expected results for the known secreted (RARP-2) and non-secreted (BFP) control proteins, as noted by the reviewer. We note that the phospho-GSK antibody is commercially available and demonstrated by the supplier to be phospho-specific using lysates treated with/without phosphatase (see <https://www.cellsignal.com/products/primary-antibodies/phospho-gsk-3b-ser9-antibody/9336>). We further note that, were the antibody not phospho-specific, then tagged BFP should be detectable in both the GSK and phospho-GSK blots; instead, tagged BFP was only detectable in the GSK blot. Finally, we note that this assay and antibody set has been used extensively to establish whether proteins are secreted into the host cell by *Rickettsia* and other bacterial

pathogens (see the works of e.g., McCaslin *et al.* [PMID 37347192], Nock *et al.* [PMID 35285700], Sanderlin *et al.* [PMID 35727033], Lehman *et al.* [PMID 29946049], Bauler *et al.* [PMID 24443531], and Garcia *et al.* [PMID 16988240]), and we have added citations to these works to the revised manuscript.

As discussed in the manuscript, expression of these GSK-tagged constructs varied considerably. Nevertheless, using antibodies against the endogenous, untagged proteins, we were able to detect Srf secretion by immunoblotting and microscopy, even for SrfS that expressed poorly in the GSK assay (see Fig. 3b–d). Although we were unable to detect expression of GSK-tagged SrfE, we successfully generated antibodies against endogenous SrfE during review of this manuscript. Using these antibodies, we show that SrfE is secreted into the host cell in the revised manuscript (Fig. 3b,c); this result underscores the importance of using multiple assays to confirm effector secretion, especially when one assay gives inconclusive results.

Specific comment #4: Fig. 5: Immunoprecipitation of endogenous SrfD from WT *R. parkeri*-infected host cytoplasmic lysates followed by mass spectrometry was performed to identify potential binding partners of the bacterial protein in the infected host cell. This experiment identified Sec61 α and β proteins from the host cell as potential binding partners of SrfD. However, the biological relevance of this interaction remains unknown and the localisation of SrfD to the endoplasmic reticulum does not seem to be affected by its interaction with the Sec61.

Response: We agree with the reviewer's assessment of our work, and we are eager to resolve the contributions of all of these novel effectors in future studies.

Specific comment #5: Overall, the findings presented, while intriguing, necessitate further validation through additional experiments to strengthen the impact of the study. It is my belief that these additional experiments are imperative to substantiate the conclusions drawn. I appreciate the authors' contributions to the field and encourage them to consider the suggested enhancements before resubmitting to ensure the manuscript reaches its full scientific potential.

Response: We thank the reviewer for their review of our work. Since such additional validation experiments were not delineated, however, we hope our point-by-point assessment addresses the specific concerns raised above. We reiterate that we performed multiple orthogonal assays to rigorously confirm Srf secretion, going well beyond that typically seen in the field. Individually, these assays each have their own limitations, as the reviewer correctly notes. However, the combined use of these assays supports our conclusion that the SrfS identified by cell-selective BONCAT are *bona fide* secreted effectors.

REVIEWERS' COMMENTS

Reviewer #1 (Remarks to the Author):

Dear authors:

Thanks you for addressing my concerns with your report. I am very satisfied with your edits and congratulate you on an excellent work here that will substantially advance the field of rickettsial molecular biology and pathogenesis. Kudos!

Reviewer #2 (Remarks to the Author):

The revised manuscript has addressed prior concerns and is adequate for publication. Great work!

Reviewer #4 (Remarks to the Author):

This work provides valuable insights and contributes to the field by expanding the applicability of cell-selective BONCAT to the obligate intracellular pathogen *R. parkeri*. I initially had some doubts regarding the use of IGEPAL for selective lysis. Transmission electron microscopy could provide direct evidence that *R. parkeri* cells remain intact in the presence of the non-ionic detergent. Additionally, immuno-electron microscopy would be a better choice for subcellular localization experiments. Nevertheless, the results of the secreted Srf immunoblotting (Fig. 3c) and immunofluorescence assays (Fig. 3d) convinced me that Srf proteins are secreted. Therefore, I would like to ask the authors to comment on why VERO, HeLa and HEK293T cells were used instead of A549 cells for the validation experiments.

Response to Reviewers' Comments

Reviewer 1:

General comments: Dear authors: Thanks you for addressing my concerns with your report. I am very satisfied with your edits and congratulate you on an excellent work here that will substantially advance the field of rickettsial molecular biology and pathogenesis. Kudos!

Response: We thank the reviewer for their support and for their helpful suggestions to improve the quality of our manuscript.

Reviewer 2:

General comments: The revised manuscript has addressed prior concerns and is adequate for publication. Great work!

Response: We thank the reviewer for their support and for their constructive input on how to improve our manuscript.

Reviewer 4:

General comments: This work provides valuable insights and contributes to the field by expanding the applicability of cell- selective BONCAT to the obligate intracellular pathogen *R. parkeri*. I initially had some doubts regarding the use of IGEPAL for selective lysis. Transmission electron microscopy could provide direct evidence that *R. parkeri* cells remain intact in the presence of the non-ionic detergent. Additionally, immuno- electron microscopy would be a better choice for subcellular localization experiments. Nevertheless, the results of the secreted Srf immunoblotting (Fig. 3c) and immunofluorescence assays (Fig. 3d) convinced me that Srf proteins are secreted. Therefore, I would like to ask the authors to comment on why VERO, HeLa and HEK293T cells were used instead of A549 cells for the validation experiments.

Response: We thank the reviewer for their timely input and for critically assessing our work. We have clarified our use of different host cells for validation experiments in the Methods section as follows:

Lines 616–617: “Vero cells were chosen for their routine use in propagating rickettsiae and performing rickettsial GSK secretion assays.”

Lines 696–697: “HeLa cells were chosen to study exogenous Srf localization patterns over A549s due to their superior transfection efficiency.”

Lines 774–776: “HEK293T cells were chosen to study exogenous SrfD over A549s due to their superior transfection efficiency and routine use in co-immunoprecipitation experiments.”